# $\mathbf{A^2CiD^2}$: Accelerating Asynchronous Communication in Decentralized Deep Learning

**Adel Nabli**
Concordia University, Mila
Sorbonne University, ISIR, CNRS
`adel.nabli@sorbonne-universite.fr`

**Eugene Belilovsky**
Concordia University, Mila

**Edouard Oyallon**
Sorbonne University, ISIR, CNRS

## Abstract

Distributed training of Deep Learning models has been critical to many recent successes in the field. Current standard methods primarily rely on synchronous centralized algorithms which induce major communication bottlenecks and synchronization locks at scale. Decentralized asynchronous algorithms are emerging as a potential alternative but their practical applicability still lags. In order to mitigate the increase in communication cost that naturally comes with scaling the number of workers, we introduce a principled asynchronous, randomized, gossip-based optimization algorithm which works thanks to a continuous local momentum named $\mathbf{A^2CiD^2}$. Our method allows each worker to continuously process mini-batches without stopping, and run a peer-to-peer averaging routine in parallel, reducing idle time. In addition to inducing a significant communication acceleration at no cost other than adding a local momentum variable, minimal adaptation is required to incorporate $\mathbf{A^2CiD^2}$ to standard asynchronous approaches. Our theoretical analysis proves accelerated rates compared to previous asynchronous decentralized baselines and we empirically show that using our $\mathbf{A^2CiD^2}$ momentum significantly decrease communication costs in poorly connected networks. In particular, we show consistent improvement on the ImageNet dataset using up to 64 asynchronous workers (A100 GPUs) and various communication network topologies.

## 1 Introduction

As Deep Neural Networks (DNNs) and their training datasets become larger and more complex, the computational demands and the need for efficient training schemes continues to escalate. Distributed training methods offer a solution by enabling the parallel optimization of model parameters across multiple workers. Yet, many of the current distributed methods in use are synchronous, and have significantly influenced the design of cluster computing environments. Thus, both the environments and algorithms rely heavily on high synchronicity in machine computations and near-instantaneous communication in high-bandwidth networks, favoring the adoption of centralized algorithms [7].

However, several studies [27, 44, 2, 28] are challenging this paradigm, proposing decentralized asynchronous algorithms that leverage minor time-delays fluctuations between workers to enhance the parallelization of computations and communications. Unlike centralized algorithms, decentralized approaches allow each node to contribute proportionally to its available resources, eliminating the necessity for a global central worker to aggregate results. Combined with asynchronous peer-to-peer (p2p) communications, these methods can streamline the overall training

37th Conference on Neural Information Processing Systems (NeurIPS 2023).

process, mitigating common bottlenecks. This includes the Straggler Problem [42], the synchronization between computations and communications [9], or bandwidth limitations [47], potentially due to particular network topologies like a ring graph [43]. However, due to the large number of parameters which are optimized, training DNNs with these methods still critically requires a considerable amount of communication [22], presenting an additional challenge [32].

This work aims to address these challenges by introducing a principled acceleration method for pair-wise communications in peer-to-peer training of DNNs, in particular for cluster computing. While conventional synchronous settings accelerate communications by integrating a Chebychev acceleration followed by Gradient Descent steps [37], the potential of accelerated asynchronous pair-wise gossip for Deep Learning (DL) remains largely unexplored. Notably, the sophisticated theory of Stochastic Differential Equations (SDEs) offers an analytical framework for the design and study of the convergence of these algorithms [12]. We introduce a novel algorithm $\mathbf{A}^2\mathbf{CiD}^2$ (standing for **A**ccelerating **A**synchronous **C**ommunication **i**n

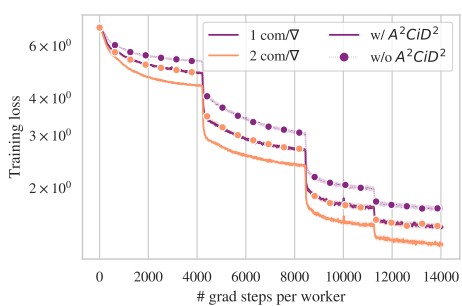

Figure 1: Adding $\mathbf{A}^2\mathbf{CiD}^2$ has the same effect as doubling the communication rates on ImageNet on the ring graph with 64 workers. See Sec. 4.

**D**ecentralized **D**eep Learning) that requires minimal overhead and effectively decouples communications and computations, accelerating pair-wise communications via a provable, accelerated, randomized gossip procedure based on continuous momentum (i.e., a mixing ODE) and time [12, 34]. We emphasize that beyond the aforementioned hardware superiority, stochastic algorithms also allows us to theoretically reach sublinear rates in convex settings [10], which opens the possibility to further principled accelerations. In practice, our method enables a virtual doubling of the communication rate in challenging network topologies without any additional cost, simply by maintaining a local momentum variable in each worker (see Fig. 1).

Our key contributions are as follows: **(1)** We extend the continuized framework [12] to the non-convex setting, in order to obtain a neat framework to describe asynchronous decentralized DL training. **(2)** This framework allows us to refine the analysis of a baseline asynchronous decentralized optimization algorithm. **(3)** We propose a novel and simple continuized momentum which allows to significantly improve communication efficiency in challenging settings, which we name $\mathbf{A}^2\mathbf{CiD}^2$. **(4)** We demonstrate that our method effectively minimizes the gap between centralized settings in environments hosting up to 64 asynchronous GPUs. **(5)** Our code is implemented in Pytorch [35], remove locks put on previous asynchronous implementations by circumventing their deadlocks, and can be found in an open-source repository: https://github.com/AdelNabli/ACiD.

This paper is structured as follows: Sec. 3.1 outlines our model for asynchronous decentralized learning, while Sec. 3.2 discusses the training dynamic used to optimize our Deep models. Sec. 3.4 offers a comprehensive theoretical analysis of our method, which is validated empirically in Sec. 4.

## 2 Related Work

**Large-scale distributed DL.** Two paradigms allow to maintain high-parallelization. On one side, model-parallelism [9, 25], which splits a neural network on independent machines, allowing to use local learning methods [4, 3]. On the other hand data-parallelism, which accelerates learning by making use of larger mini-batch splitted across multiple nodes [38] to maximally use GPU capacities. This parallelization entailing the use of larger batch-sizes, it requires an important process of adapting hyper-parameters [16], and in particular the learning rate scheduler. Developed for this setting, methods such as [16, 46] allow to stabilize training while maintaining good generalization performances. However, they have been introduced in the context of centralized synchronous training using All-Reduce schemes for communication, which still is the default setting of many approaches to data parallelism.

**Decentralized DL.** The pioneer work [27] is one of the first study to suggest the potential superiority of synchronous decentralized training strategies in practice. In terms of implementation in the cluster

setting, decentralized frameworks have been shown to achieve higher throughput than optimized All-Reduce strategies [45, 38]. From the theoretical side, [21] propose a framework covering many settings of synchronous decentralized learning. However, as it consistently relies on using a global discrete iterations count, the notion of time is more difficult to exploit, which reveals crucial in our setting. Furthermore, no communication acceleration is incorporated in these algorithms. [22] provides a comprehensive methodology to relate the consensus distance, *i.e.* the average distance of the local parameters to the global average, to a necessary communication rate to avoid degrading performance and could be easily applied to our method. [29] is a method focusing on improving performance via a discrete momentum modification, which indicates momentum variables are key to decentralized DL.

**Asynchronous Decentralized DL.** There exist many attempts to incorporate asynchrony in decentralized training [48, 5, 8, 28, 2], which typically aim at removing lock barriers of synchronous decentralized algorithms. To the best of our knowledge, none of them introduce communication acceleration, yet they could be simply combined with our approach. Although recent approaches such as [2, 28] perform peer-to-peer averaging of parameters instead of gradients, thus allowing to communicate *in parallel* of computing (as there is no need to wait for the gradients before communicating), they are still coupled: parameter updates resulting from computations and communications are scheduled in a specific order, limiting their speed. Furthermore, in practice, both those works only implement a periodic averaging on the exponential graph (more favorable, see [43]) instead of investigating the influence of the graph's topology on the convergence of a randomized gossip method, as we do. In fact, AD-PSGD [28], the baseline algorithm in asynchronous decentralized DL, comes with a major caveat to avoid deadlocks in practice: they *require* a bipartite graph and schedule p2p communications in a pseudo-random manner instead of basing the decision on worker's current availability, hindering the advantage given by asynchronous methods in the mitigation of stragglers. Contrary to them, our implementation allows to pair workers in real time based on their availability, minimizing idle time for communications.

**Communication reduction.** Reducing communication overhead is an important topic for scalability [36]. For instance, [19, 20] allow to use of compression factor in limited bandwidth setting, and the local SGD communication schedule of [30] is shown to be beneficial. Those methods could be independently and simply combined with ours to potentially benefit from an additional communication acceleration. By leveraging key properties of the resistance of the communication network [14], [12] showed that standard asynchronous gossip [6] can be accelerated, even to give efficient primal algorithms in the convex setting [34]. However, this acceleration has never been deployed in the DL context, until now. RelaySum [41] is an approach which allow to average exactly parameters produced by different time steps and thus potentially delayed. However, it requires either to use a tree graph topology, either to build ad-hoc spanning trees and has inherent synchronous locks as it averages neighbor messages in a specific order.

**Notations:** Let $n \in \mathbb{N}^*$ and $d \in \mathbb{N}^*$ an ambient dimension, for $x = (x^1, ..., x^n) \in \bigotimes_{i=1}^n \mathbb{R}^d$, we write $\bar{x} = \frac{1}{n} \sum_{i=1}^n x^i$ and $\mathbf{1}$ the tensor of ones such that $\bar{x} = \frac{1}{n} \mathbf{1}^\top x$. $\Xi$ is a probability space with measure $\mathcal{P}$. $f(t) = \mathcal{O}(1)$ means there is a $C > 0$ such that for $t$ large enough, $|f(t)| \leq C$, whereas $\tilde{\mathcal{O}}$-notation hides constants and polylogarithmic factors..

## 3 Method

### 3.1 Model for a decentralized environment

We consider a network of $n$ workers whose connectivity is given by edges $\mathcal{E}$. Local computations are modeled as (stochastic) point-wise processes $N_t^i$, and communications between nodes $(i, j) \in \mathcal{E}$ as $M_t^{ij}$. We assume that the communications are symmetric, meaning that if a message is sent from node $i$ to $j$, then the reverse is true. In practice, such processes are potentially highly correlated and could follow any specific law, and could involve delays. For the sake of simplicity, we do not model lags, though it is possible to obtain guarantees via dedicated Lyapunov functions [13]. In our setting, we assume that all nodes have similar buffer variables which correspond to a copy of a common model (e.g., a DNN). For a parameter $x$, we write $x_t^i$ the model's parameters at node $i$ and time $t$ and $x_t = (x_t^1, ..., x_t^n)$ their concatenation. In the following, we assume that each worker computes

Table 1: Comparison of convergence rates for strongly convex and non-convex objectives against concurrent works in the fixed topology setting. We neglect logarithmic terms. Observe that thanks to the maximal resistance $\chi_2 \leq \chi_1$, our method obtains substantial acceleration for the bias term. Moreover, while our baseline is strongly related to AD-PSGD [28], our analysis refines its complexity when workers sample data from the same distribution.

| Method | Strongly Convex | Non-Convex |
|---|---|---|
| Koloskova et al. [21] | $\frac{\sigma^2}{n\mu^2\epsilon} + \sqrt{L}\frac{\chi_1\xi + \sqrt{\chi_1}\sigma}{\mu^{3/2}\sqrt{\epsilon}} + \frac{L}{\mu}\chi_1$ | $\frac{L\sigma^2}{n\epsilon^2} + L\frac{\chi_1\xi + \sqrt{\chi_1}\sigma}{\epsilon^{3/2}} + \frac{L\chi_1}{\epsilon}$ |
| AD-PSGD [28] | - | $L\frac{\sigma^2+\xi^2}{\epsilon^2} + \frac{n^2 L\chi_1}{\epsilon}$ |
| Baseline **(Ours)** | $\frac{\sigma^2+\chi_1\xi^2}{\mu^2\epsilon} + \frac{L}{\mu}\chi_1$ | $L\frac{\sigma^2+\chi_1\xi^2}{\epsilon^2} + \frac{L\chi_1}{\epsilon}$ |
| $A^2CiD^2$ **(Ours)** | $\frac{\sigma^2+\sqrt{\chi_1\chi_2}\xi^2}{\mu^2\epsilon} + \frac{L}{\mu}\sqrt{\chi_1\chi_2}$ | $L\frac{\sigma^2+\sqrt{\chi_1\chi_2}\xi^2}{\epsilon^2} + \frac{L\sqrt{\chi_1\chi_2}}{\epsilon}$ |

about 1 mini-batch of gradient per unit of time (not necessarily simultaneously), which is a standard homogeneity assumption [18], and we denote by $\lambda^{ij}$ the instantaneous expected frequency of edge $(i,j)$, which we assume time homogeneous.

**Definition 3.1** (Instantaneous expected Laplacian). We define the Laplacian $\Lambda$ as:

$$\Lambda \triangleq \sum_{(i,j)\in\mathcal{E}} \lambda^{ij}(e_i - e_j)(e_i - e_j)^\mathsf{T}. \tag{1}$$

In this context, a natural quantity is the algebraic connectivity [6] given by:

$$\chi_1 \triangleq \sup_{\|x\|=1, x\perp\mathbf{1}} \frac{1}{x^\mathsf{T}\Lambda x}. \tag{2}$$

For a connected graph (*i.e.* , $\chi_1 < +\infty$), we will also use the maximal resistance of the network:

$$\chi_2 \triangleq \frac{1}{2} \sup_{(i,j)\in\mathcal{E}} (e_i - e_j)^\mathsf{T}\Lambda^+(e_i - e_j) \leq \chi_1. \tag{3}$$

The next sections will show that it is possible to accelerate the asynchronous gossip algorithms from $\chi_1$ to $\sqrt{\chi_1\chi_2} \leq \chi_1$, while [12] or [34] emphasize the superiority of accelerated asynchronous gossips over accelerated synchronous ones.

### 3.2 Training dynamic

The goal of a typical decentralized algorithm is to minimize the following quantity:

$$\inf_{x\in\mathbb{R}^d} f(x) \triangleq \inf_{x\in\mathbb{R}^d} \frac{1}{n}\sum_{i=1}^n f_i(x) = \inf_{x_i=x_1} \frac{1}{n}\sum_{i=1}^n f_i(x_i).$$

For this, we follow a first order optimization strategy consisting in using estimates of the gradient $\nabla f_i(x_i)$ via i.i.d unbiased Stochastic Gradient (SG) oracles given by $\nabla F_i(x_i, \xi_i)$ s.t. $\mathbb{E}_{\xi_i}[\nabla F_i(x_i, \xi_i)] = \nabla f_i(x_i)$. The dynamic of updates of our model evolves as the following SDE, for $\eta, \gamma, \alpha, \tilde{\alpha}$ some time-independent scalar hyper-parameters, whose values are found in our theoretical analysis and used in our implementation, and $dN_t^i(\xi_i)$ some point processes on $\mathbb{R}_+ \times \Xi$ with intensity $dt \otimes d\mathcal{P}$:

$$dx_t^i = \eta(\tilde{x}_t^i - x_t^i)dt - \gamma \int_\Xi \nabla F_i(x_t^i, \xi_i)\, dN_t^i(\xi_i) - \alpha \sum_{j,(i,j)\in\mathcal{E}} (x_t^i - x_t^j)dM_t^{ij}, \tag{4}$$

$$d\tilde{x}_t^i = \eta(x_t^i - \tilde{x}_t^i)dt - \gamma \int_\Xi \nabla F_i(x_t^i, \xi_i)\, dN_t^i(\xi_i) - \tilde{\alpha} \sum_{j,(i,j)\in\mathcal{E}} (x_t^i - x_t^j)dM_t^{ij}.$$

We emphasize that while the dynamic Eq. 4 is formulated using SDEs [1], which brings the power of the continuous-time analysis toolbox, it is still *event-based* and thus discrete in nature. Hence, it can efficiently model practically implementable algorithms, as shown by Algo. 1. The coupling $\{x_t, \tilde{x}_t\}$ corresponds to a momentum term which will be useful to obtain communication acceleration as explained in the next section. Again, $\int_{\Xi} \nabla F_i(x_t^i, \xi_i) \, dN_t^i(\xi_i)$ will be estimated via i.i.d SGs sampled as $N_t^i$ spikes. Furthermore, if $\bar{x}_0 = \bar{\tilde{x}}_0$, then, $\bar{x}_t = \bar{\tilde{x}}_t$ and we obtain a tracker of the average across workers which is similar to what is achieved through Gradient Tracking methods [19]. This is a key advantage of our method to obtain convergence guarantees, which writes as:

$$d\bar{x}_t = -\gamma \frac{1}{n} \sum_{i=1}^{n} \int_{\Xi} \nabla F_i(x_t^i, \xi_i) \, dN_t^i(\xi_i) \,. \tag{5}$$

### 3.3 Informal explanation of the dynamic through the Baseline case

To give some practical intuition on our method, we consider a baseline asynchronous decentralized dynamic, close to AD-PSGD [28]. By considering $\eta = 0, \alpha = \tilde{\alpha} = \frac{1}{2}$, the dynamic (4) simplifies to:

$$dx_t^i = -\gamma \int_{\Xi} \nabla F_i(x_t^i, \xi_i) \, dN_t^i(\xi_i) - \frac{1}{2} \sum_{j,(i,j)\in\mathcal{E}} (x_t^i - x_t^j) dM_t^{ij} \,. \tag{6}$$

In a DL setting, $x^i$ contains the parameters of the DNN hosted on worker $i$. Thus, (6) simply says that the parameters of the DNN are updated either by taking local SGD steps, or by pairwise averaging with peers $j, (i,j) \in \mathcal{E}$. These updates happen independently, at random times: although we assume that all workers compute gradients at the same speed *on average* (and re-normalized time accordingly), the use of Poisson Processes model the inherent variability in the time between these updates. However, the p2p averaging depends on the capabilities of the network, and we allow each link $(i,j)$ to have a different bandwidth, albeit constant through time, modeled through the frequency $\lambda^{ij}$. The gradient and communication processes are decoupled: there is no need for one to wait for the other, allowing to compute stochastic gradients uninterruptedly and run the p2p averaging in parallel, as illustrated by Fig.2. Finally, (4) adds a momentum step mixing the local parameters $x^i$ and momentum buffer $\tilde{x}^i$ before each type of update, allowing for significant savings in communication costs, as we show next.

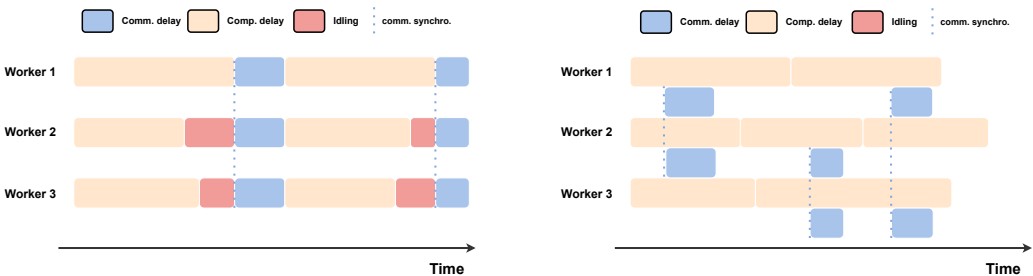

Figure 2: Example of worker updates in synchronous (**left**) and asynchronous (**right**) optimization methods. We remark that our asynchronous algorithm reduces idle time, and allow to communicate *in parallel* of computing gradient, only synchronizing two workers at a time for averaging parameters. Here, one p2p communication is performed per computation *in expectation*.

### 3.4 Theoretical analysis of $A^2CiD^2$

We now provide an analysis of our decentralized, asynchronous algorithm. For the sake of simplicity, we will consider that communications and gradients spike as Poisson processes:

**Assumption 3.2** (Poisson Processes). $N_t^i$, $M_t^{ij}$ are independent, Point-wise Poisson Processes. The $\{N_t^i\}_{i=1}^n$ have a rate of 1, and for $(i,j) \in \mathcal{E}$, $M_t^{ij}$ have a rate $\lambda^{ij}$.

We also assume that the communication network is connected during the training:

**Assumption 3.3** (Strong connectivity). We assume that $\chi_1 < \infty$.

We will now consider two generic assumptions obtained from [21], which allow us to specify our lemma to convex and non-convex settings. Note that the non-convex Assumption 3.5 generalizes the assumptions of [28], by taking $M = P = 0$.

**Assumption 3.4** (Strongly convex setting). Each $f_i$ is $\mu$-strongly convex and $L$-smooth, and:

$$\frac{1}{n}\sum_{i=1}^{n}\mathbb{E}_{\xi_i}[\|\nabla F_i(x,\xi_i) - \nabla f_i(x)\|^2] \leq \sigma^2 \text{ and } \frac{1}{n}\sum_{i=1}^{n}\|\nabla f_i(x^*) - \nabla f(x^*)\|^2 \leq \zeta^2.$$

**Assumption 3.5** (Non-convex setting). Each $f_i$ is $L$-smooth, and there exists $P, M > 0$ such that:

$$\forall x \in \mathbb{R}^d, \frac{1}{n}\sum_{i=1}^{n}\|\nabla f_i(x) - \nabla f(x)\|^2 \leq \zeta^2 + P\|\nabla f(x)\|^2,$$

and,

$$\forall x_1, ..., x_n \in \mathbb{R}^d, \frac{1}{n}\sum_{i=1}^{n}\mathbb{E}_{\xi_i}\|\nabla F_i(x_i,\xi_i) - \nabla f_i(x_i)\|^2 \leq \sigma^2 + \frac{M}{n}\sum_{i=1}^{n}\|\nabla f_i(x_i)\|^2.$$

We can now state our convergence guarantees. An informal way to understand our proposition, is that while gradient updates are non-convex, the communication updates are linear and thus benefit from local convexity; its proof is delayed to Appendix C.

**Proposition 3.6** (Convergence guarantees.). *Assume that $\{x_t, \tilde{x}_t\}$ follow the dynamic Eq. 4 and that Assumption 3.2-3.3 are satisfied. Assume that $\mathbf{1}\bar{x}_0 = x_0 = \tilde{x}_0$ and let $T$ the total running time. Then:*

- ***Non-accelerated setting**, we pick $\eta = 0, \alpha = \tilde{\alpha} = \frac{1}{2}$ and set $\chi = \chi_1$,*

- ***Acceleration ($A^2CiD^2$)**, we set $\eta = \frac{1}{2\sqrt{\chi_1\chi_2}}, \alpha = \frac{1}{2}, \tilde{\alpha} = \frac{1}{2}\sqrt{\frac{\chi_1}{\chi_2}}$, and $\chi = \sqrt{\chi_1\chi_2} \leq \chi_1$.*

*Then, there exists a constant step size $\gamma > 0$ such that if:*

- *(**strong-convexity**) the Assumption 3.4 is satisfied, then $\gamma \leq \frac{1}{16L(1+\chi)}$ and:*

$$\mathbb{E}\left[\|\bar{x}_T - x^*\|^2\right] = \tilde{\mathcal{O}}\left(\|\bar{x}_0 - x^*\|^2 e^{-\frac{\mu T}{16L(1+\chi)}} + \frac{\sigma^2 + \zeta^2(1+\chi)}{\mu^2 T}\right),$$

- *(**non-convexity**) the Assumption 3.5 is satisfied, then there is $c > 0$ which depends only on $P, M$ from the assumptions such that $\gamma \leq \frac{c}{L(\chi+1)}$ and:*

$$\frac{1}{T}\int_0^T \mathbb{E}\left[\|\nabla f(\bar{x}_t)\|^2\right] dt = \mathcal{O}\left(\frac{L(1+\chi)}{T}(f(x_0) - f(x^*)) + \sqrt{\frac{L(f(x_0) - f(x^*))}{T}(\sigma^2 + (1+\chi)\xi^2)}\right).$$

*Also, the expected number of gradient steps is $nT$ and the number of communications is $\frac{\text{Tr}(\Lambda)}{2}T$.*

Tab. 1 compares our convergence rates with concurrent works. Compared to every concurrent work, the bias term of $\mathbf{A^2CiD^2}$ is smaller by a factor $\sqrt{\frac{\chi_1}{\chi_2}} \geq 1$ at least. Yet, as expected, in the non-accelerated setting, we would recover similar rates to those. Compared to [20], the variance terms held no variance reduction with the number of workers; however, this should not be an issue in a DL setting, where it is well-known that variance reduction techniques degrade generalization during training [15]. Comparing directly the results of [2] is difficult as they only consider the asymptotic rate, even if the proof framework is similar to [28] and should thus lead to similar rates of convergence.

## 3.5 Informal interpretation and comparison with decentralized synchronous methods

Here, we informally discuss results from Prop. 3.6 and compare our communication rate with state-of-the-art decentralized synchronous methods such as DeTAG [31], MSDA [37] and OPAPC [23].

As we normalize time so that each node takes one gradient step per time unit in expectation, one time unit for us is analogous to one round of computation (one "step") for synchronous methods. Synchronous methods such as [31, 37, 23] perform multiple rounds of communications

Table 2: # of communications per "step"/time unit on several graphs.

| Method | Star | Ring | Complete |
|---|---|---|---|
| Accelerated Synchronous (*e.g.*, [31, 37, 23]) | $n^{3/2}$ | $n^2$ | $n^2$ |
| $\mathbf{A}^2\mathbf{CiD}^2$ | $n$ | $n^2$ | $n$ |

(their *Accelerated Gossip* procedure) between rounds of gradient computations by using an inner loop inside their main loop (the one counting "steps"), so that the graph connectivity do not impact the total number of "steps" necessary to reach $\epsilon$-precision. As Prop. 3.6 shows, the quantity $1 + \sqrt{\chi_1[\Lambda]\chi_2[\Lambda]}$ is a factor in our convergence rate. $\Lambda$ containing the information of both the topology $\mathcal{E}$ and the edge communication rates $\lambda_{ij}$, this is analogous to saying $\sqrt{\chi_1[\Lambda]\chi_2[\Lambda]} = \mathcal{O}(1)$ for our method (*i.e.*, the graph connectivity does not impact the time to converge), which, given the graph's topology, dictates the communication rate, see Appendix D for more details. Tab. 2 compares the subsequent communication rates with synchronous methods.

## 4 Numerical Experiments

Now, we experimentally compare $\mathbf{A}^2\mathbf{CiD}^2$ to a synchronous baseline All-Reduce SGD (AR-SGD, see [26]) and an *asynchronous baseline* using randomized pairwise communications (a variant of AD-PSGD [28], traditionally used in state-of-the-art decentralized asynchronous training of DNNs). In our case, the *asynchronous baseline* corresponds to the dynamic Eq. (6). Our approach is standard: we empirically study the decentralized training behavior of our asynchronous algorithm by training ResNets [17] for image recognition. Following [2], we pick a ResNet18 for CIFAR-10 [24] and ResNet50 for ImageNet [11]. To investigate how our method scales with the number of workers, we run multiple experiments using up to 64 NVIDIA A100 GPUs in a cluster with 8 A100 GPUs per node using an Omni-PAth interconnection network at 100 Gb/s, and set one worker per GPU.

### 4.1 Experimental Setup

**Hyper-parameters.** Training a DNN using multiple workers on a cluster requires several adaptations compared to the standard setting. As the effective batch-size grows linearly with the

Table 3: Training times on CIFAR10 ($\pm$ 6s).

| | $n$ | 4 | 8 | 16 | 32 | 64 |
|---|---|---|---|---|---|---|
| Ours | $t$ (min) | **20.9** | **10.5** | **5.2** | **2.7** | **1.5** |
| AR | $t$ (min) | 21.9 | 11.1 | 6.6 | 3.2 | 1.8 |

number of workers $n$, we use the learning-rate schedule for large batch training of [16] in all our experiments. Following [30], we fixed the local batch size to 128 on both CIFAR-10 and ImageNet.

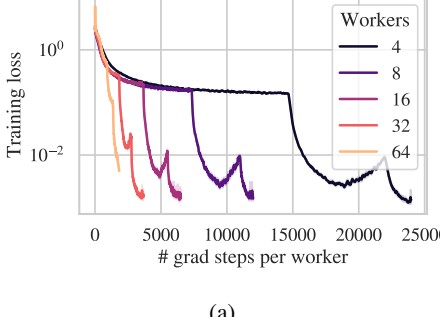

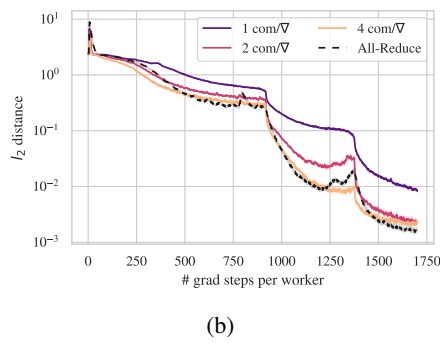

(a)                                                        (b)

Figure 3: (a) Training loss for CIFAR10 with minibatch size 128 on the complete graph, w/o $\mathbf{A}^2\mathbf{CiD}^2$. As the number of worker increases, the loss degrades, especially for $n = 64$. (b) Focus on the training loss for the complete graph of size $n = 64$, w/o $\mathbf{A}^2\mathbf{CiD}^2$. As the rate of communication increases, the gap with All-Reduce decreases. With 2 com/$\nabla$, a test accuracy of $94.6 \pm 0.04$ is reached.

Our goal being to divide the compute load between the $n$ workers, all methods access the same total amount of data samples, regardless of the number of local steps. On CIFAR-10 and ImageNet, this

**Algorithm 1:** This algorithm block describes our implementation of our Asynchronous algorithm with $\mathbf{A^2CiD^2}$ on each local machine. p2p comm. and $\nabla$ comp. are run independently in parallel.

---

**Input:** On each machine $i \in \{1, ..., n\}$, gradient oracle $\nabla f_i$, parameters $\eta, \alpha, \tilde{\alpha}, \gamma, T$.

1   **Initialize** on each machine $i \in \{1, ..., n\}$:
2     **Initialize** $x^i, \tilde{x}^i \leftarrow x^i, t^i \leftarrow 0$ and **put** $x^i, \tilde{x}^i, t^i$ in shared memory;
3     **Synchronize** the clocks of all machines ;
4   **In parallel** *on workers* $i \in \{1, ..., n\}$, **while** $t < T$, **continuously do:**
5     **In one thread** *on worker* $i$ **continuously do:**
6       $t \leftarrow clock()$ ;
7       Sample a batch of data via $\xi_i \sim \Xi$;
8       $g_i \leftarrow \nabla F_i(x_i, \xi_i)$ ;                    // Compute gradients
9       $\begin{pmatrix} x^i \\ \tilde{x}^i \end{pmatrix} \leftarrow \exp\left( (t - t^i) \begin{pmatrix} -\eta & \eta \\ \eta & -\eta \end{pmatrix} \right) \begin{pmatrix} x^i \\ \tilde{x}^i \end{pmatrix}$;
10      $x^i \leftarrow x^i - \gamma g_i$ ;                         // Apply $\mathbf{A^2CiD^2}$
11      $\tilde{x}^i \leftarrow \tilde{x}^i - \gamma g_i$ ;                      // Take the grad step
12      $t^i \leftarrow t$ ;
13     **In one thread** *on worker* $i$ **continuously do:**
14       $t \leftarrow clock()$ ;
15       Find available worker $j$ ;           // Synchronize workers $i$ and $j$
16       $m_{ij} \leftarrow (x^i - x^j)$ ;         // Send $x^i$ to $j$ and receive $x^j$ from $j$
17       $\begin{pmatrix} x^i \\ \tilde{x}^i \end{pmatrix} \leftarrow \exp\left( (t - t^i) \begin{pmatrix} -\eta & \eta \\ \eta & -\eta \end{pmatrix} \right) \begin{pmatrix} x^i \\ \tilde{x}^i \end{pmatrix}$ ;     // Apply $\mathbf{A^2CiD^2}$
18      $x^i \leftarrow x^i - \alpha m_{ij}$ ;                     // p2p averaging
19      $\tilde{x}^i \leftarrow \tilde{x}^i - \tilde{\alpha} m_{ij}$ ;
20      $t^i \leftarrow t$ ;
21   **return** $(x^i_T)_{1 \leq i \leq n}$.

---

number is set to 300 and 90 epochs respectively, following standard practice [22]. To circumvent the fuzziness of the notion of epoch in the asynchronous decentralized setting, we do not "split the dataset and re-shuffle it among workers at each epoch" as done with our standard All-Reduce baseline. Rather, we give access to the whole dataset to all workers, each one shuffling it with a different random seed. We use SGD with a base learning rate of $0.1$, a momentum value set at $0.9$ and $5 \times 10^{-4}$ for weight decay. As advocated in [16], we do not apply weight decay on the learnable batch-norm coefficients. For ImageNet training with the SGD baseline, we decay the learning-rate by a factor of 10 at epochs 30, 60, 80 (epochs 50, 75 for CIFAR-10), and apply an analogous decay schedule with our asynchronous decentralized methods. All of our neural network parameters are initialized with the default Pytorch settings, and one All-Reduce averaging is performed before and after the training to ensure consensus at initialization and before testing. For our continuous momentum, we also need to set the parameters $\eta, \tilde{\alpha}$. For all our experiments, we use the values given by Prop. 3.6. As advocated, the *asynchronous baseline* correspond to the setting without acceleration, i.e. with $\eta = 0$ and $\alpha = \tilde{\alpha} = \frac{1}{2}$, whereas using $\mathbf{A^2CiD^2}$ leads to consider $\eta = \frac{1}{2\sqrt{\chi_1 \chi_2}}, \alpha = \frac{1}{2}, \tilde{\alpha} = \frac{1}{2}\sqrt{\frac{\chi_1}{\chi_2}}$, where $\chi_1, \chi_2$ are set to their theoretical value given by (2), (3) depending on the communication rate and graph's topology, assuming that each worker chose their peers uniformly among their neighbors (we verify empirically that it is the case in practice, see Appendix E.2).

**Practical implementation of the dynamic.** The dynamic studied in Eq. (4) is a model displaying many of the properties sought after in practice. In our implementation, described in Algo. 1, each worker $i$ has indeed two independent processes and the DNN parameters and momentum variable $\{x^i, \tilde{x}^i\}$ are locally stored such that both processes can update them at any time. One process continuously performs gradient steps, while the other updates $\{x^i, \tilde{x}^i\}$ via peer-to-peer averaging. The gradient process maximizes its throughput by computing forward and backward passes back to back. Contrary to All-Reduce based methods that require an increasing number of communications with the growing number of workers, inevitably leading to an increasing time between two rounds of computations,

we study the case where each worker has a fixed communication rate, given as hyperparameter in our implementation. We implement 3 different graph topologies: complete, ring, and exponential [28, 2], see Appendix E.1 for details. To emulate the P.P.Ps for the communications, each worker samples a random number of p2p averaging to perform between each gradient computation, following a Poisson law using the communication rate as mean. To minimize idle time of the communication process, workers are paired with one of their neighbors in a "First In First Out" manner in an availability queue (a worker is available when it finished its previous averaging and still has some to do before the next gradient step). To implement this, we use a central coordinator to store the availability queues and the graph topology (this

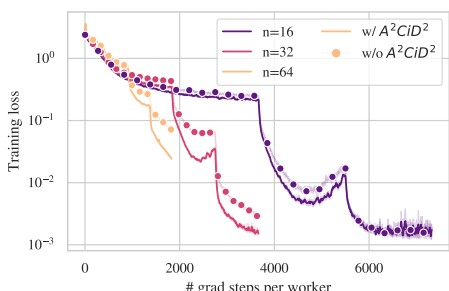

Figure 4: Training loss for CIFAR10 using a minibatch size of 128. We display the training loss with up to 64 workers, w/ and w/o $\mathbf{A}^2\mathbf{CiD}^2$, on the challenging ring graph.

is lightweight in a cluster: the coordinator only exchanges integers with the workers), but it could be done in different ways, *e.g.* by pinging each other at high frequency. As we assumed a unit time for the gradient process in our analysis, and that real time is used in our algorithm to apply our $\mathbf{A}^2\mathbf{CiD}^2$ momentum (see Algo. 1), we maintain a running average measure of the duration of the previous gradient steps to normalize time.

### 4.2 Evaluation on large scale datasets

**CIFAR10.** This simple benchmark allows to understand the benefits of our method in a well-controlled environment. Tab. 4 reports our numerical accuracy on the test set of CIFAR10, with a standard deviation calculated over 3 runs. Three scenarios are considered: a complete, an exponential and a ring graph. In Fig. 3 (a), we observe that with the asynchronous baseline on the complete graph, the more workers, the more the training loss degrades. Fig. 3 (b) hints that it is in part due to an insufficient communication rate, as increasing it allows to lower the loss and close the gap with the All-Reduce baseline. However, this is not the only causative factor as Tab. 4 indicates that accuracy generally degrades as the number of

Table 5: Accuracy on ImageNet for a batch-size of 128. We compared a vanilla asynchronous pairwise gossip approach with and without $\mathbf{A}^2\mathbf{CiD}^2$, demonstrating the improvement of our method. We also varied the communication rate.

| #Workers | #com/#grad | 16 | 32 | 64 |
|---|---|---|---|---|
| AR-SGD baseline | - | 75.5 | 75.2 | 74.5 |
| **Complete graph** Async. baseline | 1 | 74.6 | 73.8 | 71.3 |
| **Ring graph** Async. baseline | 1 | **74.8** | 71.6 | 64.1 |
| $\mathbf{A}^2\mathbf{CiD}^2$ | 1 | 74.7 | **73.4** | **68.0** |
| Async. baseline | 2 | 74.8 | 73.7 | 68.2 |
| $\mathbf{A}^2\mathbf{CiD}^2$ | 2 | **75.3** | **74.4** | **71.4** |

workers increases even for AR-SGD, which is expected for large batch sizes. Surprisingly, even with a worse training loss for $n = 64$, the asynchronous baseline still leads to better generalization than

Table 4: Accuracy of our method on CIFAR10 for a 128 batchsize with an equal number of pairwise communications and gradient computations per worker. We compared a vanilla asynchronous pairwise gossip approach with and without $\mathbf{A}^2\mathbf{CiD}^2$, demonstrating the improvement of our method.

| #Workers | 4 | 8 | 16 | 32 | 64 |
|---|---|---|---|---|---|
| AR-SGD baseline | 94.5±0.1 | 94.4±0.1 | 94.5±0.2 | 93.7±0.3 | 92.8±0.2 |
| **Complete graph** Async. baseline | 94.93±0.11 | 94.91±0.07 | 94.86±0.01 | 94.55±0.01 | 93.38±0.21 |
| **Exponential graph** Async. baseline | 95.07±0.01 | 94.89±0.01 | 94.82±0.06 | 94.44±0.02 | 93.41±0.02 |
| $\mathbf{A}^2\mathbf{CiD}^2$ | **95.17**±0.04 | **95.04**±0.01 | **94.87**±0.02 | **94.56**±0.01 | **93.47**±0.01 |
| **Ring graph** Async. baseline | **95.02**±0.06 | 95.01±0.01 | 95.00±0.01 | 93.95±0.11 | 91.90±0.10 |
| $\mathbf{A}^2\mathbf{CiD}^2$ | 94.95±0.02 | **95.01**±0.10 | **95.03**±0.01 | **94.61**±0.02 | **93.08**±0.20 |

AR-SGD, and consistently improves the test accuracy across all tested values of $n$. The communication rate being identified as a critical factor at large scale, we tested our continuous momentum on the ring graph, each worker performing one p2p averaging for each gradient step. Fig. 4 shows that incorporating $\mathbf{A}^2\mathbf{CiD}^2$ leads to a significantly better training dynamic for a large number of workers, which translates into better performances at test time as shown in Tab. 4.

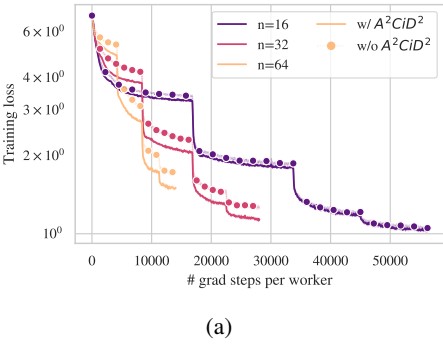
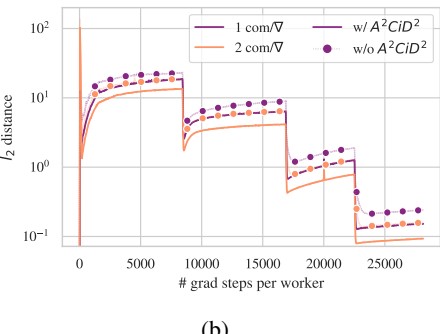

(a)                                                                                  (b)

Figure 5: (a) Training loss for ImageNet using 128 batch size, with an equal number of communications and computations per worker. We display the training loss for various number of workers (up to 64), using $\mathbf{A}^2\mathbf{CiD}^2$, for the ring graph. (b) Comparison of consensus distances when $\mathbf{A}^2\mathbf{CiD}^2$ is applied versus doubling the rate of communications on the rign graph with 64 workers: applying $\mathbf{A}^2\mathbf{CiD}^2$ has the same effect as doubling communications.

**ImageNet.** For validating our method in a real-life environment, we consider the large-scale ImageNet dataset. Tab. 6 confirms the advantage of asynchronous methods by allocating less compute to the slowest workers, leading to faster training times. Tab. 5 reports our accuracy for

Table 6: Statistics of runs on Imagenet with 64 workers (for ours, on the exponential graph).

| Method | $t$ (min) | # $\nabla$ slowest worker | # $\nabla$ fastest worker |
|---|---|---|---|
| AR-SGD | $1.7\,10^2$ | 14k | 14k |
| Baseline (ours) | $\mathbf{1.5\,10^2}$ | **13k** | 14k |
| $\mathbf{A}^2\mathbf{CiD}^2$(ours) | $\mathbf{1.5\,10^2}$ | **13k** | 14k |

the complete and ring graphs. As $\chi_1 = \chi_2$ for the complete graph, we simply run our baseline asynchronous method for reference. The case of the ring graph is much more challenging: for $n = 64$ workers, the accuracy drops by 10% compared to the synchronous baseline given by AR-SGD. Systematically, with $\mathbf{A}^2\mathbf{CiD}^2$, the final accuracy increases: up to 4% absolute percent in the difficult $n = 64$ setting. This is corroborated by Fig. 5, which indicates that incorporating $\mathbf{A}^2\mathbf{CiD}^2$ significantly improves the training dynamic on ImageNet. However, for reducing the gap with the AR-SGD baseline, it will be necessary to increase the communication rate as discussed next.

**Consensus improvement.** The bottom of Tab. 5, as well as Fig. 5 (b) study the virtual acceleration thanks to $\mathbf{A}^2\mathbf{CiD}^2$. Not only increasing communications combined with $\mathbf{A}^2\mathbf{CiD}^2$ allows to obtain competitive performance, but Fig. 1 shows that doubling the rate of communication has an identical effect on the training loss than adding $\mathbf{A}^2\mathbf{CiD}^2$. This is verified in Fig. 5 (b) by tracking the consensus distance between workers: $\mathbf{A}^2\mathbf{CiD}^2$ significantly reduces it, which validates the results of Sec. 3.4.

## 5   Conclusion

In this work, we confirmed that the communication rate is a key performance factor to successfully train DNNs at large scale with decentralized asynchronous methods. We introduced $\mathbf{A}^2\mathbf{CiD}^2$, a continuous momentum which only adds a minor local memory overhead while allowing to mitigate this need. We demonstrated, both theoretically and empirically, that $\mathbf{A}^2\mathbf{CiD}^2$ substantially improves performances, especially on challenging network topologies. As we only focused on data parallel methods for training Deep Neural Networks in a cluster environment, in a future work, we would like to extend our empirical study to more heterogeneous compute and data sources, as our theory could encompass local SGD methods [39] and data heterogeneity inherent in Federated Learning [33].

## Acknowledgements

EO, AN, and EB's work was supported by Project ANR-21-CE23-0030 ADONIS and EMERG-ADONIS from Alliance SU. This work was granted access to the HPC/AI resources of IDRIS under the allocation AD011013743 made by GENCI. EB and AN acknowledge funding and support from NSERC Discovery Grant RGPIN- 2021-04104, FRQNT New Scholar, and resources from Compute Canada and Calcul Quebec. In addition, the authors would like to thank Olexa Bilaniuk and Louis Fournier for their helpful insights regarding our code implementation.

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

# Appendix

## Table of Contents

## A  Notations

For $n \in \mathbb{N}^*$ the number of workers and $d \in \mathbb{N}^*$ an ambient dimension, for all $t > 0$, the variable $x_t \in \mathbb{R}^{n \times d}$ is a matrix such that $x_t = [x_t^1, ..., x_t^n]^\mathsf{T}$, with $x_t^i \in \mathbb{R}^d$ for all $i \in \{1, ..., n\}$. We remind that $\mathbf{1}$ is the vector of $n$ ones such that $\bar{x} = \frac{1}{n} \sum_{i=1}^n x_i = \frac{1}{n} x^\mathsf{T} \mathbf{1} \in \mathbb{R}^d$. With $\mathbf{I}$ the identity and $\|.\|_F$ the matrix Frobenius norm, we write $\pi = \mathbf{I} - \frac{1}{n} \mathbf{1} \mathbf{1}^\mathsf{T}$ the projection so that $\|\pi x\|_F^2 = \sum_{i=1}^n \|x_i - \bar{x}\|^2$.

For a continuously differentiable function $f : \mathbb{R}^d \to \mathbb{R}$ and $a, b \in \mathbb{R}^d$, the Bregman divergence is defined with $d_f(a, b) = f(a) - f(b) - \langle \nabla f(b), a - b \rangle$. For $\Lambda \in \mathbb{R}^{n \times n}$, we denote by $\Lambda^+$ its pseudo inverse. For a positive semi-definite $\Lambda \in \mathbb{R}^{n \times n}$, and $x \in \mathbb{R}^{n \times d}$, we introduce $\|x\|_\Lambda^2 \triangleq \mathrm{Tr}(x^\mathsf{T} \Lambda x)$ and $\Lambda^{1/2}$ its square-root. We recall that the connectivity between workers is given by a set of edges $\mathcal{E}$, and denote by $e_i$ the $i^{th}$ basis vector of $\mathbb{R}^n$.

For $x \in \mathbb{R}^{n \times d}$, we introduce:

$$\nabla F(x) \triangleq [\nabla f_1(x_1), ...., \nabla f_n(x_n)]^\mathsf{T} \in \mathbb{R}^{n \times d} \text{ and } \nabla \tilde{F}(x, \xi) \triangleq [\nabla F_1(x_1, \xi_1), ..., \nabla F_n(x_n, \xi_n)]^\mathsf{T} \in \mathbb{R}^{n \times d}.$$

Finally, to study the gradient steps taken on each individual worker independently, we introduce:

$$\nabla \tilde{F}_i(x, \xi) \triangleq [0, ...0, \nabla F_i(x_i, \xi_i), 0, ..., 0]^\mathsf{T} \in \mathbb{R}^{n \times d}.$$

## B  Technical Preliminaries

We recall some basic properties that we will use throughout our proofs.

**Lemma B.1** (Implications of Assumption 3.4)**.** *If each $f_i$ is $\mu$-strongly convex and $L$-smooth, we have, for any $a, b \in \mathbb{R}^d$:*

$$\frac{1}{2L} \|\nabla f_i(a) - \nabla f_i(b)\|^2 \leq d_{f_i}(a, b) \leq \frac{L}{2} \|a - b\|^2,$$

*and*

$$\frac{\mu}{2} \|a - b\|^2 \leq d_{f_i}(a, b) \leq \frac{1}{2\mu} \|\nabla f_i(a) - \nabla f_i(b)\|^2.$$

**Lemma B.2** (Generalized triangle inequality). *For any $a, b, c \in \mathbb{R}^d$ and continuously differentiable function $f : \mathbb{R}^d \to \mathbb{R}$, by definition of the Bregman divergence, we have:*

$$d_f(a, b) + d_f(b, c) = d_f(a, c) + \langle a - b, \nabla f(c) - \nabla f(b) \rangle.$$

**Lemma B.3** (Variance decomposition). *For a random vector $a \in \mathbb{R}^d$ and any $b \in \mathbb{R}^d$, the variance of $a$ can be decomposed as:*

$$\mathbb{E}\left[\|a - \mathbb{E}[a]\|^2\right] = \mathbb{E}\left[\|a - b\|^2\right] - \mathbb{E}\left[\|\mathbb{E}[a] - b\|^2\right].$$

**Lemma B.4** (Jensen's inequality). *For any vectors $a_1, ..., a_n \in \mathbb{R}^d$, we have:*

$$\left\|\sum_{i=1}^{n} a_i\right\|^2 \leq n \sum_{i=1}^{n} \|a_i\|^2.$$

**Lemma B.5.** *For any vectors $a, b \in \mathbb{R}^d$ and $\alpha > 0$:*

$$2\langle a, b \rangle \leq \alpha \|a\|^2 + \alpha^{-1} \|b\|^2.$$

**Lemma B.6.** *For any vectors $a, b \in \mathbb{R}^d$ and $\alpha > 0$:*

$$\|a - b\|^2 \leq (1 + \alpha)\|a\|^2 + (1 + \alpha^{-1})\|b\|^2.$$

**Lemma B.7.** *For any $A \in \mathbb{R}^{n \times d}$ and $B \in \mathbb{R}^{n \times n}$, we have:*

$$\|BA\|_F \leq \|A\|_F \|B\|_2.$$

**Lemma B.8** (Effective resistance contraction). *For $(i, j) \in \mathcal{E}$ and any $x \in \mathbb{R}^{n \times d}$, we have:*

$$\|(e_i - e_j)(e_i - e_j)^\mathsf{T} x\|_{\Lambda^+}^2 \leq \chi_2 \|(e_i - e_j)(e_i - e_j)^\mathsf{T} x\|_F^2.$$

*Proof.* Indeed, we note that, by definition of $\chi_2$ (3):

$$\|(e_i - e_j)(e_i - e_j)^\mathsf{T} x\|_{\Lambda^+}^2 = \mathrm{Tr}\left(x^\mathsf{T}(e_i - e_j)(e_i - e_j)^\mathsf{T} \Lambda^+ (e_i - e_j)(e_i - e_j)^\mathsf{T} x\right) \tag{7}$$

$$\leq 2\chi_2 \mathrm{Tr}\left(x^\mathsf{T}(e_i - e_j)(e_i - e_j)^\mathsf{T} x\right) \tag{8}$$

$$= \chi_2 \|(e_i - e_j)(e_i - e_j)^\mathsf{T} x\|_F^2 \tag{9}$$

$\square$

**Lemma B.9.** *For any $x \in \mathbb{R}^{n \times d}$, and $\Lambda$ the Laplacian of a connected graph, we have:*

$$\sum_{(i,j) \in \mathcal{E}} \lambda^{ij} \|(e_i - e_j)(e_i - e_j)^\mathsf{T} x\|_F^2 = 2\|x\|_\Lambda^2.$$

*Proof.* Indeed, by definition of the Laplacian $\Lambda$ (3.1), we have:

$$\sum_{(i,j) \in \mathcal{E}} \lambda^{ij} \|(e_i - e_j)(e_i - e_j)^\mathsf{T} x\|_F^2 = \sum_{(i,j) \in \mathcal{E}} \lambda^{ij} \mathrm{Tr}\left(x^\mathsf{T}(e_i - e_j)(e_i - e_j)^\mathsf{T}(e_i - e_j)(e_i - e_j)^\mathsf{T} x\right)$$

$$\tag{10}$$

$$= 2 \sum_{(i,j) \in \mathcal{E}} \lambda^{ij} \mathrm{Tr}\left(x^\mathsf{T}(e_i - e_j)(e_i - e_j)^\mathsf{T} x\right) \tag{11}$$

$$= 2\mathrm{Tr}\left(x^\mathsf{T} \sum_{(i,j) \in \mathcal{E}} \lambda^{ij}(e_i - e_j)(e_i - e_j)^\mathsf{T} x\right) \tag{12}$$

$\square$

**Lemma B.10.** *For any $x, y \in \mathbb{R}^{n \times d}$, and $\Lambda$ the Laplacian of a connected graph, we have:*

$$2\langle \pi x, y \rangle \leq \frac{1}{4}\|x\|_\Lambda^2 + 4\chi_1\|y\|_F^2$$

*Proof.* By property of the Laplacian of a connected graph, we have that $\pi = (\Lambda^+)^{1/2}\Lambda^{1/2}$. Thus:

$$2\langle \pi x, y \rangle = 2\langle \Lambda^{1/2}x, (\Lambda^+)^{1/2}y \rangle \tag{13}$$

$$\overset{\text{(B.5)}}{\leq} \frac{1}{4}\|\Lambda^{1/2}x\|_F^2 + 4\|(\Lambda^+)^{1/2}y\|_F^2 \tag{14}$$

$$\overset{\text{(B.7)}}{\leq} \frac{1}{4}\|x\|_\Lambda^2 + 4\chi_1\|y\|_F^2 \tag{15}$$

$\square$

## C    Proof of the main result of this paper

We remind that we study the following dynamic:

$$dx_t^i = \eta(\tilde{x}_t^i - x_t^i)dt - \gamma \int_\Xi \nabla F_i(x_t^i, \xi_i)\, dN_t^i(\xi_i) - \alpha \sum_{j,(i,j)\in\mathcal{E}} (x_t^i - x_t^j)dM_t^{ij},$$

$$d\tilde{x}_t^i = \eta(x_t^i - \tilde{x}_t^i)dt - \gamma \int_\Xi \nabla F_i(x_t^i, \xi_i)\, dN_t^i(\xi_i) - \tilde{\alpha} \sum_{j,(i,j)\in\mathcal{E}} (x_t^i - x_t^j)dM_t^{ij},$$

which simplifies to:

$$dx_t^i = -\gamma \int_\Xi \nabla F_i(x_t^i, \xi_i)\, dN_t^i(\xi_i) - \alpha \sum_{j,(i,j)\in\mathcal{E}} (x_t^i - x_t^j)dM_t^{ij}$$

if $\mathbf{A}^2\mathbf{CiD}^2$ is not applied. We also recall the main proposition, which we prove next:

**Proposition C.1** (Convergence guarantees.)**.** *Assume that $\{x_t, \tilde{x}_t\}$ follow the dynamic Eq. 4 and that Assumption 3.2-3.3 are satisfied. Assume that $\mathbf{1}\bar{x}_0 = x_0 = \tilde{x}_0$ and let $T$ the total running time. Then:*

- ***Non-accelerated setting***, *we pick $\eta = 0, \alpha = \tilde{\alpha} = \frac{1}{2}$ and set $\chi = \chi_1$,*

- ***Acceleration ($\mathbf{A}^2\mathbf{CiD}^2$)***, *we set $\eta = \frac{1}{2\sqrt{\chi_1\chi_2}}, \alpha = \frac{1}{2}, \tilde{\alpha} = \frac{1}{2}\sqrt{\frac{\chi_1}{\chi_2}}$, and $\chi = \sqrt{\chi_1\chi_2} \leq \chi_1$.*

*Then, there exists a constant step size $\gamma > 0$ such that if:*

- *(**strong-convexity**) the Assumption 3.4 is satisfied, then $\gamma \leq \frac{1}{16L(1+\chi)}$ and[1]:*

$$\mathbb{E}\left[\|\bar{x}_T - x^*\|^2\right] = \tilde{\mathcal{O}}\left(\|\bar{x}_0 - x^*\|^2 e^{-\frac{\mu T}{16L(1+\chi)}} + \frac{\sigma^2 + \zeta^2(1+\chi)}{\mu^2 T}\right),$$

- *(**non-convexity**) the Assumption 3.5 is satisfied, then there is $c > 0$ which depends only on $P, M$ from the assumptions such that $\gamma \leq \frac{c}{L(\chi+1)}$ and:*

$$\frac{1}{T}\int_0^T \mathbb{E}\left[\|\nabla f(\bar{x}_t)\|^2\right]\, dt = \mathcal{O}\left(\frac{L(1+\chi)}{T}(f(x_0) - f(x^*)) + \sqrt{\frac{L(f(x_0) - f(x^*))}{T}(\sigma^2 + (1+\chi)\zeta^2)}\right).$$

*Also, the expected number of gradient steps is $nT$ and the number of communications is $\frac{\text{Tr}(\Lambda)}{2}T$.*

---

[1] $\tilde{\mathcal{O}}$-notation hides constants and polylogarithmic factors.

*Proof.* The core of the proof is to introduce the appropriate Lyapunov potentials $\phi_k(t, X)$, where $X$ could be $(x, \tilde{x})$ or $x$ depending on whether we apply $\mathbf{A^2CiD^2}$ or not. If we apply $\mathbf{A^2CiD^2}$, we introduce the momentum matrix $\mathcal{A} = \begin{pmatrix} -\eta & \eta \\ \eta & -\eta \end{pmatrix}$. Then, by Ito's lemma, given that all the functions are smooth, and remembering that all Point-wise Poisson Processes $N_t^i$ have unit intensity, that the $M_t^{ij}$ have intensity $\lambda^{ij}$ and that they are all independent, we obtain in a similar fashion to [12, 34]:

$$\phi_k(T, X_T) - \phi_k(0, X_0) = \int_0^T \partial_t \phi_k(t, X_t) + \underbrace{\langle \mathcal{A} X_t, \partial_X \phi_k(t, X_t) \rangle}_{\text{momentum term}} dt$$

$$+ \sum_{i=1}^n \int_0^T \underbrace{\int_\Xi \phi_k\left(t, X_t - \gamma \begin{pmatrix} \nabla \tilde{F}_i(x_t, \xi) \\ \nabla \tilde{F}_i(x_t, \xi) \end{pmatrix}\right) - \phi_k(t, X_t)}_{\text{variation due to each independent gradient update}} dt d\mathcal{P}(\xi)$$

$$+ \sum_{(i,j) \in \mathcal{E}} \int_0^T \underbrace{\left[\phi_k\left(t, X_t - \begin{pmatrix} \alpha(e_i - e_j)(e_i - e_j)^{\mathsf{T}} x_t \\ \tilde{\alpha}(e_i - e_j)(e_i - e_j)^{\mathsf{T}} x_t \end{pmatrix}\right) - \phi_k(t, X_t)\right]}_{\text{variation due to each independent p2p communication}} \lambda^{ij} dt$$

$$+ M_T,$$

where $M_T$ is a martingale. In the case where $\mathbf{A^2CiD^2}$ is not applied, we set $\mathcal{A} = 0$ to remove the momentum term, and all updates are done only along $x$ as there is no $\tilde{x}$. We remind that:

$$\int_0^t e^{\alpha u} du = \frac{1}{\alpha}(e^{\alpha t} - 1) \tag{16}$$

We now present our choice of potential for each cases:

- For the convex case in the non-accelerated setting, we introduce:
$$\phi_1(t, x) \triangleq A_t \|\bar{x} - x^*\|^2 + B_t \|\pi x\|_F^2.$$

- For the convex case with $\mathbf{A^2CiD^2}$, we introduce:
$$\phi_2(t, x, \tilde{x}) \triangleq A_t \|\bar{x} - x^*\|^2 + B_t \|\pi x\|_F^2 + \tilde{B}_t \|\tilde{x}\|_{\Lambda^+}.$$

- For the non-convex case in the non-accelerated setting, we introduce:
$$\phi_3(t, x) \triangleq A_t d_f(\bar{x}, x^*) + B_t \|\pi x\|_F^2.$$

- For the non-convex case with $\mathbf{A^2CiD^2}$, we introduce:
$$\phi_4(t, x) \triangleq A_t d_f(\bar{x}, x^*) + B_t \|\pi x\|_F^2 + \tilde{B}_t \|\tilde{x}\|_{\Lambda^+}^2.$$

### C.1 Some useful upper-bounds

As the same terms appear in several potentials, we now prepare some intermediary results which will be helpful for the proofs.

**Study of the $\|\bar{x} - x^*\|^2$ terms:**

First, we study the variations in the $\|\bar{x} - x^*\|^2$ term appearing in $\phi_1$ and $\phi_2$. As the updates due to the communication are in the orthogonal of $\mathbf{1}$, it is only necessary to study the variations induced by the gradient steps. Thus, we define:

$$\Delta x \triangleq \sum_{i=1}^n \overline{\|x - \gamma \nabla \tilde{F}_i(x, \xi)} - x^*\|^2 - \|\bar{x} - x^*\|^2$$

We note that $\overline{\nabla \tilde{F}_i(x, \xi)} = \frac{1}{n}\nabla F_i(x_i, \xi_i)$, which, using $\sum_i \nabla f_i(x^*) = 0$, leads to:

$$\mathbb{E}_{\xi_1,\dots,\xi_n}[\Delta x] = \sum_{i=1}^{n} -\frac{2\gamma}{n}\langle \bar{x} - x^*, \nabla f_i(x_i)\rangle + \frac{\gamma^2}{n^2}\mathbb{E}_{\xi_i}[\|\nabla F_i(x_i, \xi_i)\|^2] \tag{17}$$

$$= \sum_{i=1}^{n} -\frac{2\gamma}{n}\langle \bar{x} - x^*, \nabla f_i(x_i) - \nabla f_i(x^*)\rangle + \frac{\gamma^2}{n^2}\mathbb{E}_{\xi_i}[\|\nabla F_i(x_i, \xi_i)\|^2] \tag{18}$$

$$\overset{(3.4),(B.3)}{\leq} \frac{\gamma^2}{n}\sigma^2 + \sum_{i=1}^{n} -\frac{2\gamma}{n}\langle \bar{x} - x^*, \nabla f_i(x_i) - \nabla f_i(x^*)\rangle + \frac{\gamma^2}{n^2}\|\nabla f_i(x_i)\|^2 \tag{19}$$

$$\overset{(B.2)}{=} \frac{\gamma^2}{n}\sigma^2 + \sum_{i=1}^{n} -\frac{2\gamma}{n}(d_{f_i}(\bar{x}, x^*) + d_{f_i}(x^*, x_i) - d_{f_i}(\bar{x}, x_i)) + \frac{\gamma^2}{n^2}\|\nabla f_i(x_i)\|^2$$
$$\tag{20}$$

$$\overset{(B.4)}{\leq} \frac{\gamma^2}{n}\sigma^2 + \sum_{i=1}^{n} -\frac{2\gamma}{n}(d_{f_i}(\bar{x}, x^*) + d_{f_i}(x^*, x_i) - d_{f_i}(\bar{x}, x_i))$$
$$+ \frac{2\gamma^2}{n^2}\|\nabla f_i(x^*) - \nabla f_i(x_i)\|^2 + \frac{2\gamma^2}{n^2}\|\nabla f_i(x^*)\|^2 \tag{21}$$

$$\overset{(3.4),(B.1)}{\leq} \frac{\gamma^2}{n}\sigma^2 + \frac{2\gamma^2}{n}\zeta^2 + \sum_{i=1}^{n} -\frac{2\gamma}{n}(d_{f_i}(\bar{x}, x^*) + d_{f_i}(x^*, x_i) - d_{f_i}(\bar{x}, x_i)) + \frac{4L\gamma^2}{n^2}d_{f_i}(x^*, x_i)$$
$$\tag{22}$$

$$\overset{(B.1)}{\leq} \frac{\gamma^2\sigma^2}{n} + \frac{2\gamma^2}{n}\zeta^2 - \gamma\mu\|\bar{x} - x^*\|^2 + \frac{L\gamma}{n}\|\pi x\|_F^2 + \sum_{i=1}^{n}\left(-\frac{2\gamma}{n} + \frac{4L\gamma^2}{n^2}\right)d_{f_i}(x^*, x_i) \tag{23}$$

**Study of the $d_f(\bar{x}, x^*)$ terms:**

Next, using the same reasoning as for $\Delta x$, we also need to only study the gradient updates in the non-convex setting for the first part of $\phi_3$, $\phi_4$. Thus, we set:

$$\Delta f \triangleq \sum_{i=1}^{n} d_f\left(\bar{x} - \gamma\frac{1}{n}\nabla F_i(x_i, \xi_i), x^*\right) - d_f(\bar{x}, x^*). \tag{24}$$

First, it is useful to note that under Assumption 3.5, using (B.3), we have:

$$\mathbb{E}_{\xi_1,\dots,\xi_n}\|\nabla\tilde{F}(x, \xi)\|_F^2 \leq n\sigma^2 + (1 + M)\sum_{i=1}^{n}\|\nabla f_i(x_i)\|^2. \tag{25}$$

Then, using $\sum_i \nabla f_i(x^*) = 0$ and the $L$-smoothness of $f = \frac{1}{n}\sum_i f_i$, we get:

$$\mathbb{E}_{\xi_1,\dots,\xi_n}[\Delta f] \overset{(B.2)}{=} \sum_{i=1}^{n}\mathbb{E}[d_f(\bar{x} - \gamma\frac{1}{n}\nabla F_i(x_i, \xi_i), \bar{x})] - \frac{\gamma}{n}\langle\nabla f_i(x_i), \nabla f(\bar{x})\rangle \tag{26}$$

$$\overset{(B.1)}{\leq} \sum_{i=1}^{n}\frac{1}{2n^2}L\gamma^2\mathbb{E}[\|\nabla F_i(x_i, \xi_i)\|^2] - \frac{\gamma}{n}\langle\nabla f_i(x_i), \nabla f(\bar{x})\rangle \tag{27}$$

$$\overset{(25)}{\leq} \frac{L\gamma^2}{2n}\sigma^2 - \gamma\|\nabla f(\bar{x})\|^2 + \sum_{i=1}^{n}\frac{M+1}{2n^2}L\gamma^2\|\nabla f_i(x_i)\|^2 - \sum_{i=1}^{n}\frac{\gamma}{n}\langle\nabla f_i(x_i) - \nabla f_i(\bar{x}), \nabla f(\bar{x})\rangle$$
$$\tag{28}$$

$$\overset{(B.5)}{\leq} \frac{L\gamma^2}{2n}\sigma^2 + \frac{\gamma}{2n}L^2\|\pi x\|_F^2 - \frac{\gamma}{2}\|\nabla f(\bar{x})\|^2 + \sum_{i=1}^{n}\frac{M+1}{2n^2}L\gamma^2\|\nabla f_i(x_i)\|^2 \tag{29}$$

As we also have:

$$\sum_{i=1}^{n} \|\nabla f_i(x_i)\|^2 \overset{(B.4)}{\leq} \sum_{i=1}^{n} 3(\|\nabla f_i(x_i) - \nabla f_i(\bar{x})\|^2 + \|\nabla f_i(\bar{x}) - \nabla f(\bar{x})\|^2 + \|\nabla f(\bar{x})\|^2) \quad (30)$$

$$\overset{(3.5)}{\leq} 3L^2\|\pi x\|_F^2 + 3n\zeta^2 + 3n(1+P)\|\nabla f(\bar{x})\|^2 \quad (31)$$

We get in the end:

$$\mathbb{E}_{\xi_1,\dots,\xi_n}[\Delta f] \leq \frac{L\gamma^2}{2n}\sigma^2 + \frac{3(M+1)}{2n}\gamma^2 L\zeta^2 + \left(\frac{3(M+1)}{2n^2}L^3\gamma^2 + \frac{\gamma}{2n}L^2\right)\|\pi x\|_F^2$$
$$+ \left(\frac{3(M+1)}{2n}(1+P)L\gamma^2 - \frac{\gamma}{2}\right)\|\nabla f(\bar{x})\|^2 \quad (32)$$

*Remark* C.2. As observed in (5), note that for both the terms $\|\bar{x} - x^*\|^2$ and $d_f(\bar{x}, x^*)$, as we are considering $\bar{x}$ and $\frac{1}{n}\mathbf{1}\mathbf{1}^\mathsf{T}\pi = 0$, Poisson updates from the communication process amount to zero. Moreover, as $\frac{1}{n}\mathbf{1}\mathbf{1}^\mathsf{T}(x - \tilde{x}) = 0$, the update from the momentum is also null for these terms.

**Study of the $\|\pi x\|_F^2$ terms:**

We get from the Poisson updates for the gradient processes:

$$\mathbb{E}_{\xi_1,\dots,\xi_n}\left[\sum_{i=1}^{n} \|\pi(x - \gamma\nabla\tilde{F}_i(x,\xi))\|_F^2 - \|\pi x\|_F^2\right] = -2\gamma\langle\pi x, \nabla F(x)\rangle + \gamma^2\sum_{i=1}^{n}\mathbb{E}_{\xi_1,\dots,\xi_n}\|\pi\nabla\tilde{F}_i(x,\xi)\|_F^2$$
$$\leq -2\gamma\langle\pi x, \nabla F(x)\rangle + \gamma^2\mathbb{E}_{\xi_1,\dots,\xi_n}\|\nabla\tilde{F}(x,\xi)\|_F^2,$$

and, using the definition of the Laplacian $\Lambda$ (3.1), we get from the communication processes:

$$\sum_{(i,j)\in\mathcal{E}} \lambda^{ij}\left(\|\pi(x - \alpha(e_i - e_j)(e_i - e_j)^\mathsf{T}x)\|_F^2 - \|\pi x\|_F^2\right) = -2\alpha\langle x, \Lambda x\rangle + \sum_{(i,j)\in\mathcal{E}} \lambda^{ij}\alpha^2\|(e_i - e_j)(e_i - e_j)^\mathsf{T}x\|_F^2$$
$$\overset{(B.9)}{=} -2\alpha\langle x, \Lambda x\rangle + 2\alpha^2\|x\|_\Lambda^2$$
$$= 2\alpha(\alpha - 1)\|x\|_\Lambda^2.$$

Putting together both types of Poisson updates, we define:

$$\mathbb{E}[\Delta_\pi] \triangleq -2\gamma\langle\pi x, \nabla F(x)\rangle + \gamma^2\mathbb{E}_{\xi_1,\dots,\xi_n}\|\pi\nabla\tilde{F}(x,\xi)\|_F^2 + 2\alpha(\alpha - 1)\|x\|_\Lambda^2 \quad (33)$$

$$\overset{(B.10)}{\leq} 4\chi_1\gamma^2\|\nabla F(x)\|_F^2 + \left(\frac{1}{4} - 2\alpha(1-\alpha)\right)\|x\|_\Lambda^2 + \gamma^2\mathbb{E}_{\xi_1,\dots,\xi_n}\|\nabla\tilde{F}(x,\xi)\|_F^2 \quad (34)$$

For $\alpha = \frac{1}{2}$, we get:

$$\mathbb{E}[\Delta_\pi] \leq 4\chi_1\gamma^2\|\nabla F(x)\|_F^2 - \frac{1}{4\chi_1}\|\pi x\|_F^2 + \gamma^2\mathbb{E}_{\xi_1,\dots,\xi_n}\|\nabla\tilde{F}(x,\xi)\|_F^2 \quad (35)$$

For $\mathbf{A^2CiD^2}$, we add the momentum term $\langle\eta(\tilde{x} - x), 2\pi x\rangle$ to define:

$$\Delta_\Lambda^1 \triangleq 2\eta\langle\tilde{x}, \pi x\rangle - 2\eta\|\pi x\|_F^2 - 2\gamma\langle\pi x, \nabla F(x)\rangle + \gamma^2\mathbb{E}_{\xi_1,\dots,\xi_n}\|\nabla\tilde{F}(x,\xi)\|_F^2 + 2\alpha(\alpha - 1)\|x\|_\Lambda^2 \quad (36)$$

$$\overset{(B.5)}{\leq} 2\eta\langle\tilde{x}, \pi x\rangle - \frac{3}{2}\eta\|\pi x\|_F^2 + \frac{2}{\eta}\gamma^2\|\nabla F(x)\|_F^2 - 2\alpha(1-\alpha)\|x\|_\Lambda + \gamma^2\mathbb{E}_{\xi_1,\dots,\xi_n}\|\nabla\tilde{F}(x,\xi)\|_F^2. \quad (37)$$

**Study of the $\|\tilde{x}\|_{\Lambda^+}^2$ terms:**

These terms only appear in the Lyapunov potentials used when applying $\mathbf{A^2CiD^2}$. From the Poisson updates and momentum, we get:

$$\Delta_\Lambda^2 \triangleq 2\eta\langle x - \tilde{x}, \tilde{x}\rangle_{\Lambda^+} - 2\gamma\langle \tilde{x}, \Lambda^+\nabla\tilde{F}(x,\xi)\rangle + \gamma^2\|\nabla\tilde{F}(x,\xi)\|_{\Lambda^+}^2 - 2\tilde{\alpha}\langle\pi x, \tilde{x}\rangle$$
$$+ \tilde{\alpha}^2\sum_{(i,j)\in\mathcal{E}}\lambda^{ij}\|(e_i - e_j)(e_i - e_j)^\mathsf{T}x\|_{\Lambda^+}^2 . \tag{38}$$

Taking the expectation and using (B.9), (B.8), (B.7), (B.5) leads to:

$$\mathbb{E}[\Delta_\Lambda^2] \leq \chi_1\eta\|\pi x\|_F^2 + \left(\frac{\eta}{2} - \eta\right)\|\tilde{x}\|_{\Lambda^+}^2 + \frac{2}{\eta}\chi_1\gamma^2\|\pi\nabla F(x)\|_F^2 - 2\tilde{\alpha}\langle\pi x, \tilde{x}\rangle$$
$$+ 2\chi_2\tilde{\alpha}^2\|x\|_\Lambda + \chi_1\gamma^2\mathbb{E}_{\xi_1,\dots,\xi_n}\|\nabla\tilde{F}(x,\xi)\|_F^2 \tag{39}$$

## C.2  Resolution: putting everything together

In this part, we combine the terms for each potential. We remind that with Assumption 3.4:

$$\sum_{i=1}^n \mathbb{E}_{\xi_i}[\|\nabla F_i(x_i,\xi_i)\|^2] = n\sigma^2 + \sum_{i=1}^n \|\nabla f_i(x_i)\|^2 \tag{40}$$

$$\leq n\sigma^2 + \sum_{i=1}^n 2\|\nabla f_i(x^*) - \nabla f_i(x_i)\|^2 + 2\|\nabla f_i(x^*)\|^2 \tag{41}$$

$$\leq n\sigma^2 + 2n\zeta^2 + 4L\sum_{i=1}^n d_{f_i}(x^*, x_i) \tag{42}$$

**Convex case, non-accelerated.**  We remind that
$$\phi_1(t, x, \tilde{x}) = A_t\|\bar{x} - x^*\|^2 + B_t\|\pi x\|^2.$$
Then, using (23), (35) and defining $\Psi_1 \triangleq \partial_t\phi_1(t, X_t) + \mathbb{E}[A_t\Delta x + B_t\Delta_\pi]$, we have:

$$\Psi_1 \leq \|\bar{x} - x^*\|^2\left(A_t' - \mu\gamma A_t\right) \tag{43}$$

$$+ \|\pi x\|^2\left(B_t' + \frac{L\gamma}{n}A_t - \frac{1}{4\chi_1}B_t\right) \tag{44}$$

$$+ \sum_{i=1}^n d_{f_i}(x^*, x_i)\left(-\frac{2\gamma}{n}A_t + \frac{4L\gamma^2}{n^2}A_t + 4L\left(4\chi_1\gamma^2 + \gamma^2\right)B_t\right) \tag{45}$$

$$+ \left(\frac{\gamma^2\sigma^2}{n} + \frac{2\gamma^2}{n}\zeta^2\right)A_t + \left(n\gamma^2\sigma^2 + 2n\zeta^2(4\chi_1\gamma^2 + \gamma^2)\right)B_t \tag{46}$$

We pick $\alpha = \frac{1}{2}$, $B_t = \frac{1}{n}A_t$, with $A_t = e^{-rt}$ (we denote by $r$ the rate of the exponentials $A_t, B_t$). Then (43), (44) imply:

$$r \leq \min\left(\mu\gamma, \frac{1}{4\chi_1} - L\gamma\right) \tag{47}$$

As we want (45) to be negative, we have:

$$-1 + \gamma\left(\frac{2L}{n} + 2L(4\chi_1 + 1)\right) \leq 0 \tag{48}$$

which leads to:

$$\gamma \leq \frac{1}{2L\left(\frac{1}{n} + 4\chi_1 + 1\right)} \tag{49}$$

and taking $\gamma \leq \frac{1}{2}\frac{1}{2L(3+4\chi_1+1)} = \frac{1}{16L(1+\chi_1)}$ works. Now, as $\frac{1}{4\chi_1} - L\gamma \geq \frac{3}{16\chi_1}$ and $\mu\gamma \leq \frac{1}{16\chi_1}$, we pick $r = \mu\gamma$. As we have:

$$\mathbb{E}\left[\phi_1(T, x_T) - \phi_1(0, x_0)\right] = \int_0^T \left(A_t'\|\bar{x}_t - x^*\|^2 + B_t'\|\pi x_t\|^2 + A_t\mathbb{E}_{\xi_1,\dots,\xi_n}[\Delta x] + B_t\mathbb{E}[\Delta_\pi]\right)dt \tag{50}$$

using (46) and (16) leads to:

$$\mathbb{E}\|\bar{x}_t - x^*\|^2 \leq e^{-\gamma\mu t}\left(\|\bar{x}_0 - x^*\|^2 + \frac{1}{n}\|\pi x_0\|^2\right) + \frac{\gamma}{\mu}\left(\sigma^2(\frac{1}{n} + 1) + 2\zeta^2(\frac{1}{n} + 4\chi_1 + 1)\right) \tag{51}$$

**Convex case with A$^2$CiD$^2$.** We remind that
$$\phi_2(t, x, \tilde{x}) = A_t\|\bar{x} - x^*\|^2 + B_t\|\pi x\|^2 + \tilde{B}_t\|\tilde{x}\|_{\Lambda^+}.$$
Then, using (23), (37), (39) and defining $\Psi_2 \triangleq \partial_t\phi_2(t, X_t) + \mathbb{E}[A_t\Delta x + B_t\Delta^1_\Lambda + \tilde{B}_t\Delta^2_\Lambda]$, we have:

$$\Psi_2 \leq \|\bar{x} - x^*\|^2 \left(A'_t - \mu\gamma A_t\right) \tag{52}$$

$$+ \|\pi x\|^2 \left(B'_t + \frac{L\gamma}{n}A_t - \frac{3}{2}\eta B_t + \eta\chi_1\tilde{B}_t\right) \tag{53}$$

$$+ \|\tilde{x}\|^2_{\Lambda^+} \left(\tilde{B}'_t - \frac{\eta}{2}\tilde{B}_t\right) \tag{54}$$

$$+ \|x\|^2_\Lambda \left(2\tilde{\alpha}^2\chi_2\tilde{B}_t - 2\alpha(1 - \alpha)B_t\right) \tag{55}$$

$$+ \langle\tilde{x}, \pi x\rangle \left(2\eta B_t - 2\tilde{\alpha}\tilde{B}_t\right) \tag{56}$$

$$+ \sum_{i=1}^{n} d_{f_i}(x^*, x_i) \left(-\frac{2\gamma}{n}A_t + \frac{4L\gamma^2}{n^2}A_t + 4L\left(\frac{2\gamma^2}{\eta} + \gamma^2\right)(B_t + \chi_1\tilde{B}_t)\right) \tag{57}$$

$$+ \left(\frac{\gamma^2\sigma^2}{n} + \frac{2\gamma^2}{n}\zeta^2\right)A_t + \left(n\gamma^2\sigma^2 + 2n\zeta^2(\gamma^2 + \frac{2\gamma^2}{\eta})\right)(B_t + \chi_1\tilde{B}_t) \tag{58}$$

Then, we assume $\alpha = \frac{1}{2}$, $\tilde{\alpha} = \frac{1}{2}\sqrt{\frac{\chi_1}{\chi_2}}$, $\eta = \frac{1}{2\sqrt{\chi_1\chi_2}}$, $B_t = \frac{1}{n}A_t$, $\tilde{B}_t = \frac{1}{\chi_1}B_t$, $A_t = e^{-rt}$ (we denote by $r$ the rate of the exponentials $A_t, B_t, \tilde{B}_t$), which satisfies (55) and (56). Then (52), (53), (54) imply:

$$r \leq \min(\mu\gamma, \frac{\eta}{2}, \frac{\eta}{2} - L\gamma) = \min(\mu\gamma, \frac{\eta}{2} - L\gamma) \tag{59}$$

As we want (57) to be negative, we have:

$$-1 + \gamma\left(\frac{2L}{n} + 4L(\frac{2}{\eta} + 1)\right) \leq 0 \tag{60}$$

which leads to:

$$\gamma \leq \frac{1}{2L\left(\frac{1}{n} + \frac{4}{\eta} + 2\right)} \tag{61}$$

and taking $\gamma \leq \frac{1}{2L(6 + \frac{4}{\eta} + 2)} = \frac{1}{16L(1 + \sqrt{\chi_1\chi_2})}$ works. Now, we have:

$$\frac{\eta}{2} - L\gamma \geq \frac{1}{4\sqrt{\chi_1\chi_2}}\left(1 - \frac{4\sqrt{\chi_1\chi_2}}{16(1 + \sqrt{\chi_1\chi_2})}\right) \geq \frac{3}{16\sqrt{\chi_1\chi_2}} \tag{62}$$

As $\mu\gamma \leq \frac{\mu}{16L(1 + \sqrt{\chi_1\chi_2})} \leq \frac{1}{16\sqrt{\chi_1\chi_2}}$, taking $r = \mu\gamma$ works. Finally, using (58) and (16) leads to:

$$\mathbb{E}\|\bar{x}_t - x^*\|^2 \leq e^{-\gamma\mu t}\left(\|\bar{x}_0 - x^*\|^2 + \frac{2}{n}\|\pi x_0\|^2\right) + \frac{\gamma}{\mu}\left(\sigma^2(\frac{1}{n} + 2) + 2\zeta^2(\frac{1}{n} + 8\sqrt{\chi_1\chi_2} + 2)\right) \tag{63}$$

**Non-convex case, non-accelerated.** We remind that:
$$\phi_3(t, x) = A_t d_f(\bar{x}, x^*) + B_t\|\pi x\|^2$$

Here, we pick $\alpha = \frac{1}{2}$, $A_t = 1$, $B_t = \frac{L}{n}A_t$. Thus, $A'_t = B'_t = 0$. Then, using (32), (35), (31), (25) we obtain:

$$A_t\mathbb{E}[\Delta f] + B_t\mathbb{E}[\Delta_\pi]) \leq \|\nabla f(\bar{x})\|^2\left(-\frac{\gamma}{2}A_t + \frac{3}{2n}L\gamma^2(M + 1)(P + 1)A_t + 3n\gamma^2(4\chi_1 + M + 1)(P + 1)B_t\right) \tag{64}$$

$$+ \|\pi x\|^2\left(\frac{L^2\gamma}{2n}(1 + \frac{3}{n}(M + 1)L\gamma)A_t + 3L^2\gamma^2(4\chi_1 + M + 1)B_t - \frac{1}{4\chi_1}B_t\right) \tag{65}$$

$$+ \gamma^2\left(\sigma^2 + 3(M + 1)\zeta^2\right)\left(\frac{L}{2n}A_t + nB_t\right) + 12n\chi_1\gamma^2\zeta^2 B_t \tag{66}$$

Our goal is to use half of the negative term of (64) to cancel the positive ones, so that there remains at least $-\frac{\gamma}{4}A_t\|\nabla f(\bar{x})\|^2$ in the end. Thus, we want:

$$3L\gamma^2\left(\frac{(M+1)(P+1)}{2n}+(4\chi_1+M+1)(P+1)\right)\leq\frac{\gamma}{4} \qquad (67)$$

and taking $\gamma\leq\frac{1}{48(M+1)(P+1)(1+\chi_1)}$ works. We verify that with $\gamma$ defined as such, (65) is also negative. Finally, we upper bound (66) with $3L\gamma^2\left(\sigma^2+3(M+1+4\chi_1)\zeta^2\right)$. As we have:

$$\mathbb{E}[\phi_3(T,x_T)-\phi_3(0,x_0)]=\int_0^T\left(A_t'd_f(\bar{x}_t,x^*)+B_t'\|\pi x_t\|^2+A_t\mathbb{E}[\Delta f]+B_t\mathbb{E}[\Delta_\pi]\right)dt \qquad (68)$$

we note that if $\gamma\leq\frac{c}{L(\chi_1+1)}$ for some constant $c>0$ which depends on $M,P$, we will get:

$$\frac{\gamma}{4}\int_0^T\mathbb{E}[\|\nabla f(\bar{x}_t)\|^2]\,dt\leq\frac{L}{n}\|\pi x_0\|^2+d_f(x_0,x^*)+\mathcal{O}\left(LT\gamma^2(\sigma^2+(1+\chi_1)\zeta^2)\right) \qquad (69)$$

which also writes:

$$\frac{1}{T}\int_0^T\mathbb{E}[\|\nabla f(\bar{x}_t)\|^2]\,dt\leq\frac{4}{\gamma T}(f(x_0)-f(x^*))+\mathcal{O}\left(L\gamma(\sigma^2+(1+\chi_1)\zeta^2)\right) \qquad (70)$$

**Non-convex case, with $\mathbf{A^2CiD^2}$.** We have:

$$\phi_4(t,x)=A_td_f(\bar{x},x^*)+B_t\|\pi x\|^2+\tilde{B}_t\|\tilde{x}\|_{\Lambda+}^2$$

Here, we pick $\alpha=\frac{1}{2},\eta=\frac{1}{2\sqrt{\chi_1\chi_2}},A_t=1,B_t=\frac{L}{n}A_t,B_t=\chi_1\tilde{B}_t$ and an identical reasoning to the convex setting allows to say we can find a constant $c>0$ such that if $\gamma\leq\frac{c}{L(1+\sqrt{\chi_1\chi_2})}$, then:

$$\frac{1}{T}\int_0^T\mathbb{E}[\|\nabla f(\bar{x}_t)\|^2]\,dt=\mathcal{O}\left(\frac{1}{\gamma T}(f(x_0)-f(x^*))+L\gamma(\sigma^2+(1+\sqrt{\chi_1\chi_2})\zeta^2)\right) \qquad (71)$$

### C.3 Optimizing the step-size

In this part, we follow [40, 21] and optimize the step-size a posteriori. We set $\chi=\chi_1$ for the non-accelerated setting and $\chi=\sqrt{\chi_1\chi_2}$ with $\mathbf{A^2CiD^2}$.

**Strongly-convex cases:** From (51) and (63), we can write that, for $\gamma\leq\frac{1}{16L(1+\chi)}$ and initializing $x_0$ such that $\pi x_0=0$, we have:

$$\mathbb{E}\|\bar{x}_t-x^*\|^2=\mathcal{O}\left(\|\bar{x}_0-x^*\|^2e^{-\gamma\mu t}+\frac{\gamma}{\mu}(\sigma^2+\zeta^2(1+\chi))\right) \qquad (72)$$

Then, taking the proof of [40] and adapting the threshold, we consider two cases (with $r_0\triangleq\|\bar{x}_0-x^*\|^2$):

- if $\frac{1}{16L(1+\chi)}\geq\frac{\log(\max\{2,\mu^2r_0T/\sigma^2\})}{\mu T}$, then we set $\gamma=\frac{\log(\max\{2,\mu^2r_0T/\sigma^2\})}{\mu T}$.

  In this case, (72) gives:

$$\mathbb{E}\|\bar{x}_T-x^*\|^2=\tilde{\mathcal{O}}\left(\frac{1}{\mu^2T}(\sigma^2+\zeta^2(1+\chi))\right) \qquad (73)$$

- if $\frac{1}{16L(1+\chi)}<\frac{\log(\max\{2,\mu^2r_0T/\sigma^2\})}{\mu T}$, then we set $\gamma=\frac{1}{16L(1+\chi)}$.

  Then, (72) gives:

$$\mathbb{E}\|\bar{x}_T-x^*\|^2=\mathcal{O}\left(r_0e^{-\frac{\mu T}{16L(1+\chi)}}+\frac{1}{\mu}\frac{1}{16L(1+\chi)}(\sigma^2+\zeta^2(1+\chi))\right) \qquad (74)$$

$$=\tilde{\mathcal{O}}\left(r_0e^{-\frac{\mu T}{16L(1+\chi)}}+\frac{1}{\mu^2T}(\sigma^2+\zeta^2(1+\chi))\right) \qquad (75)$$

**Non-convex cases:** From (70) and (71), we can write that, for some constant $c > 0$ depending on $M, P$ such that $\gamma \leq \frac{c}{L(1+\chi)}$, we have:

$$\frac{1}{T} \int_0^T \mathbb{E}[\|\nabla f(\bar{x}_t)\|^2] \, dt = \mathcal{O}\left( \frac{1}{\gamma T}(f(x_0) - f(x^*)) + L\gamma(\sigma^2 + (1+\chi)\zeta^2) \right) \qquad (76)$$

Then, taking the proof of Lemma 17 in [21] and adapting the threshold, we consider two cases (with $f_0 \triangleq f(x_0) - f(x^*)$):

- if $\frac{c}{L(1+\chi)} < \left( \frac{f_0}{TL(\sigma^2 + (1+\chi)\zeta^2)} \right)^{1/2}$, then we take $\gamma = \frac{c}{L(1+\chi)}$, giving:

$$\frac{1}{T} \int_0^T \mathbb{E}[\|\nabla f(\bar{x}_t)\|^2] \, dt = \mathcal{O}\left( \frac{L(1+\chi)}{T} f_0 + L \left( \frac{f_0}{TL(\sigma^2 + (1+\chi)\zeta^2)} \right)^{1/2} (\sigma^2 + (1+\chi)\zeta^2) \right)$$

$$(77)$$

$$= \mathcal{O}\left( \frac{L(1+\chi)}{T} f_0 + \sqrt{\frac{Lf_0}{T}(\sigma^2 + (1+\chi)\zeta^2)} \right) \qquad (78)$$

- if $\frac{c}{L(1+\chi)} \geq \left( \frac{f_0}{TL(\sigma^2 + (1+\chi)\zeta^2)} \right)^{1/2}$, then we take $\gamma = \left( \frac{f_0}{TL(\sigma^2 + (1+\chi)\zeta^2)} \right)^{1/2}$, giving:

$$\frac{1}{T} \int_0^T \mathbb{E}[\|\nabla f(\bar{x}_t)\|^2] \, dt = \mathcal{O}\left( \frac{f_0}{T} \left( \frac{TL(\sigma^2 + (1+\chi)\zeta^2)}{f_0} \right)^{1/2} + \sqrt{\frac{Lf_0}{T}(\sigma^2 + (1+\chi)\zeta^2)} \right)$$

$$(79)$$

$$= \mathcal{O}\left( \sqrt{\frac{Lf_0}{T}(\sigma^2 + (1+\chi)\zeta^2)} \right) \qquad (80)$$

$\square$

## D  Comparison with accelerated synchronous methods

By definition of $\Lambda$ (3.1), our communication complexity (the expected number of communications) is simply given by $\frac{\text{Tr}(\Lambda)}{2}$ per time unit. As discussed in Sec. 3.5, our goal is to replicate the behaviour of accelerated synchronous methods such as DeTAG [31], MSDA [37] and OPAPC [23] by communicating sufficiently so that the graph connectivity does not impact the time to converge, leading to the condition $\sqrt{\chi_1[\Lambda]\chi_2[\Lambda]} = \mathcal{O}(1)$.

Now, let us consider a gossip matrix $W$ as in [31, 37, 23] (*i.e.*, $W$ is symmetric doubly stochastic) and its Laplacian $\mathcal{L} = I_n - W$. Then, using $\Lambda = \sqrt{\chi_1[\mathcal{L}]\chi_2[\mathcal{L}]}\mathcal{L}$ is sufficient for having $\sqrt{\chi_1[\Lambda]\chi_2[\Lambda]} = \mathcal{O}(1)$.

- **Synchronous methods:** between two rounds of computations ("steps"), the number of communication edges used is $\frac{|\mathcal{E}|}{\sqrt{1-\theta}}$ with $\theta = \max\{|\lambda_2|, |\lambda_n|\}$ the eigenvalues of $W$.

- **Ours:** the number of communication edges used per time unit for our method is $\frac{\text{Tr}(\Lambda)}{2} = \frac{1}{2}\sqrt{\chi_1[\mathcal{L}]\chi_2[\mathcal{L}]}\text{Tr}(\mathcal{L})$.

As, in [31, 37, 23], each communication edge is used at the same rate, we can apply **Lemma 3.3** of [34] stating: $\sqrt{\chi_1[\mathcal{L}]\chi_2[\mathcal{L}]}\text{Tr}(\mathcal{L}) \leq \sqrt{\|\mathcal{L}\|\chi_1(n-1)|\mathcal{E}|}$. We have:

- $W$ is stochastic: $\|\mathcal{L}\| \leq 2$.
- the graph is connected: $n - 1 \leq |\mathcal{E}|$.
- definition of $\chi_1$ and $\theta$: $1 - \theta \leq \frac{1}{\chi_1[\mathcal{L}]}$

Thus, $\sqrt{\chi_1[\mathcal{L}]\chi_2[\mathcal{L}]}\text{Tr}(\mathcal{L}) \leq \frac{\sqrt{2}|\mathcal{E}|}{\sqrt{1-\theta}}$, which proves that our communication complexity per time unit is at least as good as any accelerated synchronous method.

# E Experimental details

## E.1 Graph topologies

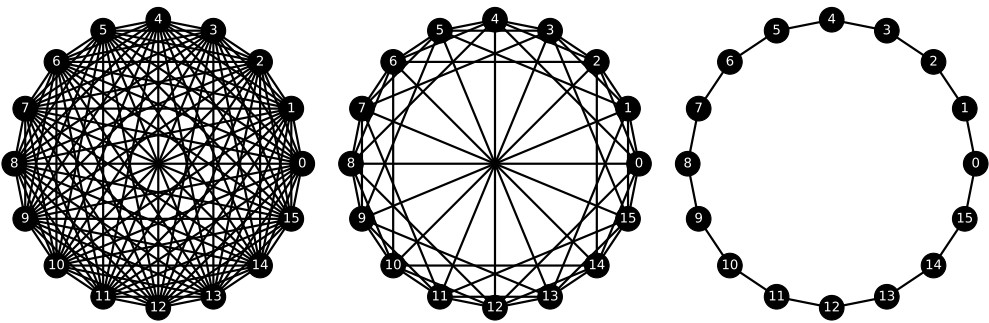

Figure 6: The three types of graph topology implemented. From left to right: complete, exponential, cycle, all with 16 nodes. From left to right, the approximate values of $(\chi_1, \chi_2)$ with a communication rate of "1 p2p comm./$\nabla$ comp." for each worker are: $(1, 1)$, $(2, 1)$, $(13, 1)$.

Fig. 6 displays an example of each of the three graph topologies implemented. The exponential graph follows the architecture described in [28, 2]. Note the discrepancy between the values of $\chi_1$ and $\chi_2$ for the cycle graph, highlighting the advantage of using $\mathbf{A}^2\mathbf{CiD}^2$ in the asynchronous setting (to lower the complexity from $\chi_1$ to $\sqrt{\chi_1\chi_2}$).

## E.2 Uniform neighbor selection check



Figure 7: Heat-map of the communication history (showed through a weighted adjacency matrix) during the asynchronous training on CIFAR10 with 32 workers. We display the results for the complete graph (left), exponential (centre) and ring (right) graph.

Our asynchronous algorithm acts as follows: to reduce latency, the first two workers (*i.e.*, GPUs) in the whole pool that declare they are ready to communicate (*i.e.*, finished their previous communication and have to perform more before the next gradient step) are paired together for a p2p communication if they are neighbors in the connectivity network. During training, we registered the history of the pairwise communications that happened. Fig. 7 displays the heat-map of the adjacency matrix, confirming that our assumption of "uniform pairing among neighbors" (used to compute the values of $\chi_1, \chi_2$) seems to be sound.

