# OpenReview forum: "$\textbf{A}^2\textbf{CiD}^2$: Accelerating Asynchronous Communication in Decentralized Deep Learning"
_NeurIPS.cc/2023/Conference — NeurIPS 2023 poster_

### Official Review · Reviewer_LKYk · 2023-06-29

**Soundness:** 2 fair
**Presentation:** 2 fair
**Contribution:** 2 fair
**Rating:** 5
**Confidence:** 4

**Summary:**

This work discusses the challenges and potential solutions related to training complex Deep Neural Networks (DNNs), particularly regarding the computational and communication demands. The traditional synchronous, centralized approaches to DNN training, while widely used, face limitations in terms of efficiency and scalability, which can be tackled by distributed training methods.

Asynchronous and decentralized methods are suggested as they allow for more efficient parallelization of computations and communications, using time-delay fluctuations between workers. Such methods eliminate the need for a central worker to aggregate results, allow nodes to contribute in proportion to their available resources, and use peer-to-peer communication to streamline training. However, the complexity of these methods and the large number of parameters in DNNs still pose considerable communication challenges.

The authors address these issues by introducing a novel acceleration method, A2CiD2 (Accelerating Asynchronous Communication in Decentralized Deep Learning), specifically for peer-to-peer DNN training. This method uses pair-wise gossip acceleration, which is largely unexplored for deep learning, and is supported by the analytical framework of Stochastic Differential Equations (SDEs). It decouples computations and communications, requiring minimal overhead and enhancing communication rates.

Key contributions of this work include extending the asynchronous decentralized deep learning training framework to non-convex settings, proposing the A2CiD2 mechanism for improving communication efficiency, and minimizing the gap between centralized and decentralized settings in environments with up to 64 asynchronous GPUs. The method has been implemented in PyTorch and will be released as open-source upon publication.

**Strengths:**

The strengths of this paper are as follows:

1: Within the realm of deep learning, the widespread use of High-Performance Computing (HPC) technology has made it possible to achieve exceptional performance in a synchronous and centralized environment. However, not everyone has the means to afford such costly setups, making the development and evaluation of distributed asynchronous algorithms crucial. The authors propose a cost-effective method that significantly enhances communication speed.

2: The A2CiD2 algorithm effectively minimizes the discrepancy between centralized and distributed setups. It functions efficiently in environments hosting up to 64 asynchronous GPUs. This characteristic enhances the algorithm's scalability, rendering it suitable for large-scale machine learning tasks.

3: This study broadens the analytical framework for investigating the design and convergence of these algorithms in non-convex settings. It provides new insights into asynchronous distributed deep learning training.

4: The method proposed substantially improves communication efficiency in distributed learning environments. This could assist in addressing common challenges, such as the straggler problem, synchronization between computation and communication, and bandwidth limitations.

**Weaknesses:**

The weaknesses of this paper can be outlined as follows:

1: While the theoretical analysis provided in Section 3.3 is well substantiated, its tightness with respect to the upper bound remains ambiguous. Further explanation is needed on how the asymptotic convergence corresponds with the experimental results for relatively smaller sizes, like those discussed in Section 4.

2: The study mentions that the proposed A2CiD2 algorithm has been tested with up to 64 asynchronous GPUs. While this is quite remarkable, it may not comprehensively represent the broad array of hardware configurations and scales found in real-world applications. A broader evaluation of the method across different scales and architectures would be advantageous.

3: The paper lacks a clear comparison with other existing distributed training methods, be they synchronous or asynchronous, in terms of computational cost, communication overhead, and performance. Such a comparison could help determine whether the proposed method truly surpasses other methods and under which specific conditions.

4: The paper asserts that training Deep Neural Networks (DNNs) with decentralized methods necessitates substantial communications due to the large quantity of optimized parameters. It remains unclear whether the proposed acceleration method sufficiently addresses this issue and how it compares with other techniques that aim to reduce communication overhead.

**Questions:**

In addition to the issues delineated as weaknesses above, I would be grateful if you could respond to the following queries:

1: In Section 3.1, "Model for a Decentralized Environment," you presuppose the symmetry of communication. Could you elaborate on this assumption and explicate its implications for your model?

2: In Section 3.3, "Theoretical Analysis of A2CiD2," you propose a Poisson process. Nevertheless, the validity of this assumption remains nebulous. Could you furnish some justification or explanation for this assumption?

3: In Section 4, "Numerical Experiments," you cite the utilization of 64 A100 GPUs. Could you provide more details about the network configuration they are connected to, such as Infiniband? The performance of the network itself is likely to affect the results of this experiment, and this information would help elucidate the experimental setup.

**Limitations:**

N/A : No pertinent content is found in the paper.

---

> ### Author Rebuttal · Authors · 2023-08-10
>
> We thank reviewer LKYk for highlighting the need for cost-effective methods enhancing communication speed in distributed training, and recognizing  that our method is a step towards rendering large-scale training possible in this setting by substantially improving  communication efficiency while providing new insight into asynchronous distributed deep learning training.
>
> **Weaknesses:**
>
> 1) Lower bounds to minimize the sum of functions with randomized algorithms exist [ [1]( https://arxiv.org/pdf/1605.08003.pdf ), [2](https://arxiv.org/pdf/1805.10222.pdf) ] and we are probably not tight in that sense. However, we still provide a SOTA communication complexity, even compared to accelerated *synchronous* methods that use Chebychev acceleration as our rate depends on the maximal effective resistance of the graph instead of the spectral gap (see, e.g. [[3]]( http://proceedings.mlr.press/v202/nabli23a/nabli23a.pdf ) ). Note that, if we put $f=0$, then the problem reduces to satisfying the consensus constraint and our algorithm meets the fastest known rate for gossip algorithms [[7]]( https://proceedings.neurips.cc/paper/2021/file/ec26fc2eb2b75aece19c70392dc744c2-Paper.pdf ). Finally, remark that convergence rates in the smooth case are often put as a sanity check in the deep learning literature as they are vacuous in practice [[8]]( http://proceedings.mlr.press/v97/arora19a/arora19a.pdf ).
>
> 2) We must stress that we work in **academia** with a **publicly funded cluster**. Thus, while we agree that verifying experimentally that our method is amenable to a wide range of hardware settings would be of interest, this is reasonably beyond the reach of the material capacity of our academic setting. Note that despite those constraints, our numerical experiments use a number of workers of the order of  state-of-the-art work [[9]]( http://proceedings.mlr.press/v97/assran19a/assran19a.pdf ). The open-source release of our code is planned to allow such experiments for other actors with more compute resources. In terms of network architecture, we will aso report additional experiments with Vision Transformers shortly (experiments still ongoing at the time of writing), but we expect a similar behavior for these models.
>
> 3) We aim at reducing the communication cost of **asynchronous** methods, because they have the potential to be faster than synchronous ones due to the removal of wait barriers. While bringing many practical advantages in the large-scale distributed setting, asynchronous algorithms also lead to specific technical challenges compared to synchronous approaches. As such, the closest method to which we can reasonably compare is AD-PSGD [ [4] ]( http://proceedings.mlr.press/v80/lian18a/lian18a.pdf ), which we have done. We stress that no other existing decentralized asynchronous method displays accelerated rates of communications in the training of neural networks: AD-PSGD [ [4] ]( http://proceedings.mlr.press/v80/lian18a/lian18a.pdf ) has a complexity depending on the *spectral gap* $\omega ,$ whereas ours depends on $\sqrt{\chi_1 \chi_2}$ which is better (typically for the cycle graph, $\omega = \mathcal O (n^2)$ and $\sqrt{\chi_1 \chi_2} = \mathcal O (n)$).
>
> 4) We believe that any method taken alone would probably not be sufficient to grow to arbitrarily scale and that our method is another tool to reduce the communication cost for the decentralized asynchronous training of neural networks. As we use a local momentum, our method is completely orthogonal to schemes aiming at reducing the bandwidth requirements through compressed or quantized communications, which can be added on top of $A^2CiD^2$.
>
> **Questions:**
>
> 1) The symmetry of communications means that our communication network is undirected: when an edge $(i,j)$ “spikes”, both nodes $i$ and $j$ send their parameters to the other, such that both can compute the *average* of $(x_i, x_j)$. While our *implementation* of our method is indeed symmetric for simplicity’s sake, what our theoretical analysis says is a bit more subtle: we only require that the *directed* and *expected* rates of communications between any two workers ($i$ to $j$ and $j$ to $i$) are the same. Thus, our algorithm could also work with non-symmetric communication schemes (i.e push or pull methods). We stress that, in practice, symmetric communications are often assumed for *asynchronous* schemes (see, e.g. [ [4] ]( http://proceedings.mlr.press/v80/lian18a/lian18a.pdf ) ).
>
> 2) The Poisson modeling of gossip algorithms is standard in decentralized peer-to-peer networks *(see, e.g., the seminal paper of Boyd et al. [[5]](https://web.stanford.edu/~boyd/papers/pdf/gossip.pdf) )*, we thus follow this literature. To gain more insight, this modeling consider that any individual link in the network has a fixed bandwidth, thus the time between  two communications is always *roughly* the same, and variations are taken into account by making this delay stochastic (following an exponential law). However, we agree that this model is not perfect, and some refinements have been made to move closer to reality (see, e.g. [ [6]](https://arxiv.org/pdf/2106.03585.pdf) ).
>
> 3) We will add the following details to the paper: all our experiments are done on a cluster with 8 A100 per node using an Omni-PAth interconnection network 100 Gb/s. In all our experiments, one worker amounts to one GPU (and not one node).

---

> > ### Comment · Reviewer_LKYk · 2023-08-16
> >
> > Thank you for your prompt responses, despite the limited time frame.
> >
> > I grasped the main points from the replies, but they didn't address my questions. It would have been helpful had you clearly outlined the limitations from the outset. I found there to be more limitations than I had initially anticipated.
> >
> > The asymptotic behavior of the algorithm for very large data sizes and its practical behavior for small, limited data sizes should be discussed systematically. The dominant terms might vary depending on whether the data size is large or small. While it's reasonable to assume that the algorithm's theoretical properties contribute to its positive experimental outcomes, various factors influence the calculations.
> >
> > Regarding Weakness 2, the objective isn't to conduct numerous experiments across multiple architectures, but to derive a broader spectrum of insights and predictions with a limited set of experiments in various environments. This topic is frequently broached in fields like HPC (High Performance Computing), making such discussions quite beneficial. If there's a model addressing computation, data, and communication quantities, a more comprehensive discussion and set of findings could emerge.
> >
> > I understand the challenge of presenting extensive information within a paper of limited length. However, after revisiting the experimental section, it seems, from a reader's perspective, to be primarily a performance evaluation within a constrained setting. Nonetheless, I'm not disputing the value of your method.

---

> > > ### Author Response · Authors · 2023-08-17
> > >
> > > Thank you very much for participating in the discussion.
> > >
> > > * There seems to be some misunderstanding about the theoretical analysis. *We don't adopt an asymptotic approach*, as we are not examining the scenario where "t" tends towards positive infinity. Instead, we focus on a quantitative approach, considering specific time steps “t”. Additionally, it's important to note that *the sizes of the data/datasets do not influence or factor into our convergence analysis*. We believe our algorithm's theoretical attributes are well mirrored by our experimental outcomes. If the AC allows us to, we will illustrate this match between our theoretical convergence rate and the observed one with a Figure (this is not surprising given that communications here are "convex," capitalizing on the corresponding theory). We kindly ask the reviewer to clarify what additional factors precisely they believe could be better addressed or clarified in the theoretical analysis.
> > > * We’re slightly unclear on the expectations of the reviewer, as our work is built upon the standard norms in the community, both from the theoretical point of view (see [1,2,3,4,5,7], even for the Poisson Process use), and from the experimental set-up perspective ([5,6,7,8] leading to our implementation using 64 GPUs, ResNet model, CIFAR10/ImageNet datasets with the objective to maintain performance under the decentralized constraints). It would be very helpful if the reviewer could provide specific references.
> > >
> > > We thank you in advance for any further clarifications.
> > >
> > > [1] *Optimal Algorithms for Smooth and Strongly Convex Distributed Optimization in Networks*, ICML 2017
> > >
> > > [2] *Optimal and Practical Algorithms for Smooth and Strongly Convex Decentralized Optimization*, NeurIPS 2020.
> > >
> > > [3] *DADAO: Decoupled Accelerated Decentralized Asynchronous Optimization*, ICML 2023.
> > >
> > > [4] *A Unified Theory of Decentralized SGD with Changing Topology and Local Updates*, ICML 2020
> > >
> > > [5] *Stochastic Gradient Push for Distributed Deep Learning*, ICML 2019
> > >
> > > [6] *Don’t use large mini-batches, use local SGD*, ICLR 2020
> > >
> > > [7] *Asynchronous Decentralized Parallel Stochastic Gradient Descent*, ICML 2018
> > >
> > > [8] *Consensus control for decentralized deep learning*, ICML 2021

---

> > > > ### Comment · Reviewer_LKYk · 2023-08-21
> > > >
> > > > Thank you for your reply.
> > > > As a result of the discussion so far, I would like to raise my rating by one step, but I still think the biggest problem with this paper is the lack of clarity in the structure of the paper.
> > > > As I have already pointed out, I think the connection between the theoretical and experimental analyses is poor. There are many gaps between the theoretical and experimental results, and I think it would be better to increase the experimental part even if the amount of the theoretical analysis part is reduced. I think it would have been better to conduct separate measurements for the calculation part and the communication part, not only to show the overall superiority of the method but also to go into the characteristics and reasons for the superiority of the method.
> > > > I don't think there are any particular problems with the choice of the computing environment or data set. Again, the lack of experimentation and discussion is problematic.

---

> > > > > ### Author Response · Authors · 2023-08-21
> > > > >
> > > > > We greatly appreciate your decision to increase the score by 1 and we thank you for acknowledging that there is no problem with our computing and datasets choices.
> > > > >
> > > > > We have produced an additional analysis (which we have shared with the AC) that empirically compares standard gossip with $A^2CiD^2$ solely from a communication standpoint, demonstrating a strong match between theoretical communication rates and empirical consensus convergence speed.
> > > > >
> > > > > We also want to re-emphasize that our experiments include  training DNNs, using one of the most challenging graph (a ring graph), conducted on 64 workers, on Imagenet, which is strong evidence of the benefits of our method from an experimental perspective. Given the theory, demonstration of its match to practice, and the evaluations in challenging settings we kindly urge the reviewer to reconsider their evaluation.
> > > > >
> > > > > Thanks for your attention.

---

### Official Review · Reviewer_fQ1m · 2023-07-04

**Soundness:** 2 fair
**Presentation:** 1 poor
**Contribution:** 2 fair
**Rating:** 4
**Confidence:** 2

**Summary:**

This work, A2CiD2, proposes a novel decentralized asynchronous training method that incurs only minimal overhead but effectively decouples communications and computations, accelerating pair-wise communications via a provable, accelerated, randomized gossip procedure based on continuous momentum and time.  A2CiD2 is also compatible with other asynchronous approaches. Across ResNet benchmarks on image classification, A2CiD2 is able to outperform AllReduce-SGD and AD-PSGD on a cluster of 64 asynchronous workers
with A100 GPUs, using various communication network topologies.

**Strengths:**

+. Proposed a decentralized asynchronous training algorithm that outperforms the SOTA approaches

+. Demonstrated the advantage of the proposed approach both theoretically and empirically

+. Provided code for reproducibility

**Weaknesses:**

-. Writing needs improvement. Hard to follow.

-. Missing benchmark:
1. only ResNet is evaluated (e.g., ResNet18 and ResNet50), how about more models like RNNs and larger models like GPT2?

2. only image classification tasks is evaluated, how about more tasks like language modeling?

-. Missing details in experiments:
1. what is the network type and bandwidth?
2. how many GPUs per machine?

-. Missing evaluation:
1. what is the time-to-accuracy or time-to-loss speedup for AC2CiD2 over AllReduce-SGD and ADPSGD?

2. what is performance of AC2CiD2 with random stranger in the cluster?

**Questions:**

See Weakeness

---

> ### Author Rebuttal · Authors · 2023-08-10
>
> We thank reviewer fQ1m for stressing that our method outperforms SOTA approaches, both theoretically and empirically. We emphasize that this paper is **not** about implementations tricks but rather about *fundamental research* on new strategies to speedup asynchronous training. Indeed, our work solves the important question of understanding how to accelerate communications in an asynchronous setting.
>
> **Weaknesses:**
>
> * We corrected a few typos and drafting errors in the revised version of our paper to clarify the writing. To mention a few, a spell-checker only found about 20 of those located at lines independent *(l36: communications -> communication), (l45: gossips->gossip),(l56: allow to->allows us to), (l58:  communication -> the communication), (l62: improve -> improves) (l66: time of -> the time of), (l73: independant -> independent), (l94: exists -> exist), (l103: to use strong -> to use of ), (l135: Next sections -> The next sections), (l159: allow to -> allow us to), (l176: step -> steps), (l186: technique degrade accurracy -> techniques degrade accuracy), (l199: adaptation -> adaptations), (l210: networks parameters-> network parameters), (l233: each others -> each other), (l238: allows -> allows us), (l280: finally-> final), (l281: the Fig. -> Fig.)* However, we believe this minor weakness does not diminish the significance of our scientific contribution.
>
> * For the image classification task, we will aso report additional experiments with Vision Transformers shortly (experiments still ongoing at the time of writing), but we expect a similar behavior for these models.We emphasize that our evaluation is standard in the literature (see [ [1](https://arxiv.org/pdf/1808.07217.pdf), [2](https://arxiv.org/pdf/1811.10792.pdf) , [3]( http://proceedings.mlr.press/v80/lian18a/lian18a.pdf ) ]). Moreover, we are unfortunately not able to train large language models such as GPT2 for lack of the right computing infrastructures. The open-source release of our code is planned to allow such experiments for other actors with more compute resources.
>
> * We will add the following details to the paper: all our experiments are done on an **academic** cluster with 8 A100 per node using an Omni-PAth interconnection network 100 Gb/s. In all our experiments, one worker amounts to one GPU (and not one node).
>
> * Thank you for raising interest on these metrics, we will provide additional figures in the revised version of our paper. In the meantime, we report some preliminary figures *(in the pdf attached to the global rebuttal)* showing that applying $A^2CiD^2$ is indeed worthwhile in terms of time.
>
> * We are not sure we understand the question. In the “complete graph” setting, our asynchronous algorithm acts as follows: to reduce latency, the first two workers (i.e., GPUs) in the whole pool that declare they are ready to communicate (i.e, finished their previous communication) are paired together for a p2p communication. In practice, as each worker has to perform a random number (following a poisson law) of communications between two gradient computations, this means that the pairs are completely random (no pair of workers can repeatedly synchronize at the same time). We verify that indeed, during the course of a training run, each edge in the complete graph appeared roughly the same amount of time *(see the pdf attached to the global rebuttal)*. We’d be happy to provide further explanations if required.
>
> While seemingly working in the High Performance Computing setting, we must stress that our experiments are mainly done to **simulate** a decentralized network (e.g, connected through the internet) and to highlight the potential of using $A^2CiD^2$ to reduce the communication cost in situations where it is a major bottleneck to train neural networks.

---

> > ### Comment · Reviewer_fQ1m · 2023-08-12
> >
> > Thanks for the rebuttal with a detailed explanation. The authors have addressed my concerns to some extent through the response, so I will raise the score by one level.

---

> > > ### Author Response · Authors · 2023-08-15
> > >
> > > Thank you for participating in the discussion!
> > >
> > > While we recognize the significance of LLMs in this context (which, we acknowledge, is a crucial concern though not yet a standard and widely established baseline in distributed literature, see e.g. [ [1](https://arxiv.org/pdf/1808.07217.pdf), [2]( http://proceedings.mlr.press/v80/lian18a/lian18a.pdf ) ]), we would like to inquire if you perceive any other gaps or areas that might require further elaboration. We’d be happy to offer additional explanations and insights as needed. Thanks.

---

### Official Review · Reviewer_o6Xd · 2023-07-10

**Soundness:** 3 good
**Presentation:** 2 fair
**Contribution:** 3 good
**Rating:** 5
**Confidence:** 4

**Summary:**

This paper proposes an asynchronous gossip-based algorithm for decentralized deep learning by using a continuous momentum. Experiments on real datasets are used for evaluation.

**Strengths:**

1. The studied problem about accelerating communication in decentralized deep learning is interesting.

2. The proposed algorithm accelerates the asynchronous baseline both theoretically and empirically.

**Weaknesses:**

1. The writing can be improved. There are many typos and grammatical errors. Furthermore, there are many informal representations, such as “We demonstrate its efficiency theoretically and numerically; empirically on the ring graph…” in abstract.

2. The proposed method needs to store one more copy of the model, compared with baselines. Hence, it may not be suitable for large models.

3. The experimental results are not convincing. The main baseline for comparison is All-Reduce SGD which has very high communication cost. There have existed many sophisticated decentralized methods, which should be adopted for comparison.

4. This work only considers undirected network topology, which means that the communications are symmetric. But in recent years, directed topology has attracted more and more attention.

**Questions:**

Is the formula between Line 139 and Line 140 correct?

**Limitations:**

yes

---

> ### Author Rebuttal · Authors · 2023-08-10
>
> We are glad that reviewer o6Xd finds the problem we study interesting and recognizes that our method accelerates previous algorithms both theoretically and empirically. We stress that our method is especially catered for large scale training of deep neural networks: the advantage of **asynchronous** algorithms grows with scale, and further accelerating them is not trivial.
>
> **Weaknesses:**
>
> 1) We acknowledge that a few typos and drafting errors have slipped through our fingers, which we will correct in the final version of this work.
> To mention a few, a spell-checker only found about 20 of those located at lines independent *(l36: communications -> communication), (l45: gossips->gossip),(l56: allow to->allows us to), (l58:  communication -> the communication), (l62: improve -> improves) (l66: time of -> the time of), (l73: independant -> independent), (l94: exists -> exist), (l103: to use strong -> to use of ), (l135: Next sections -> The next sections), (l159: allow to -> allow us to), (l176: step -> steps), (l186: technique degrade accurracy -> techniques degrade accuracy), (l199: adaptation -> adaptations), (l210: networks parameters-> network parameters), (l233: each others -> each other), (l238: allows -> allows us), (l280: finally-> final), (l281: the Fig. -> Fig.).*
> However, we believe this minor weakness does not diminish the significance of our scientific contribution.
> We clarify the abstract as follows: *"Our theoretical analysis proves accelerated rates compared to previous asynchronous decentralized baselines and we empirically show that adding $A^2CiD^2$ has the same effect as doubling the communication rate on the ring graph."*
>
> 2) We respectfully disagree with this statement. First, remark that *any* algorithm that uses SGD with momentum is also doubling the number of parameters in memory: that is the cost of storing the momentum variable in the optimizer. Thus, our method is completely analogous to a standard momentum in that sense.
> Plus, in large models such as Transformers, the main memory cost does not reside in the storing of the parameters, but rather in the memory requirements for storing the activations and computing the gradients (see e.g,  [ [ 1, ]( https://arxiv.org/pdf/2111.11124.pdf ) [ 2 ]( https://arxiv.org/pdf/2205.05198.pdf ) ] ). As our method only uses the second set of parameters as a *"memory bank"* (our second "model" does not perform any computation: neither forward nor backward), it is virtually at no cost in terms of memory (the memory for activations in the original model dominating the storing of a second set of parameters).
>
> 3) The main baseline for comparison we use is AD-PSGD [ [3] ]( http://proceedings.mlr.press/v80/lian18a/lian18a.pdf ): this is our “asynchronous baseline” in Tab.2 and Tab.4, which is quite standard and challenging to beat. Indeed, our primary objective is to minimize the communication overhead of asynchronous techniques, which are faster than synchronous methods as they aim at removing wait barriers from their implementation. However, we introduce All-Reduce SGD as a benchmark, establishing a target performance that an effective (asynchronous or synchronous/centralized or decentralized) method should achieve.While bringing many practical advantages in the large-scale distributed setting, asynchronous algorithms also lead to specific technical challenges compared to synchronous approaches. We stress that no other existing decentralized asynchronous method displays accelerated rates of communications in the training of neural networks and that any other scheme to reduce communication cost (e.g., compressed or quantized communications) can be added on top of our method. If we missed some, do you have in mind other **decentralized asynchronous** baselines to which we can compare ?
>
> 4) We respectfully disagree: using asymmetric communication can be simply done in our case. Indeed, while the *implementation* of our method is indeed symmetric for simplicity’s sake, what our theoretical analysis says is more subtle: we only require that the *directed* and *expected* rates of communications between any two workers ($i$ to $j$ and $j$ to $i$) are the same. Thus, our algorithm could also work with non-symmetric communication schemes (i.e push or pull methods). We stress that, in practice, symmetric communications are often assumed for *asynchronous* schemes (see, e.g. [ [3] ]( http://proceedings.mlr.press/v80/lian18a/lian18a.pdf ) ).
>
> **Questions:**
>
> The  formula is indeed correct, it is a very standard re-formulation of the decentralized optimization problem using the consensus constraint (see e.g eq. 10 in [ [4] ](https://arxiv.org/pdf/1702.08704.pdf) or eq. 2 in [ [5]](https://proceedings.neurips.cc/paper_files/paper/2020/file/d530d454337fb09964237fecb4bea6ce-Paper.pdf )). We’d be happy to provide further explanations if needed.

---

> > ### Comment · Area_Chair_vxtw · 2023-08-15
> > **How is the theoretical communication cost compared to the synchronous algorithm**
> >
> > Given that all workers are with the same computation time, the asynchronous case reduces to the synchronous case. Then the question is how the theoretical communication cost is compared to the synchronous case [1] which gives the optimal communication bound for decentralized algorithms.
> >
> > [1] Optimal Complexity in Decentralized Training, ICML 2021

---

> > > ### Author Response · Authors · 2023-08-15
> > >
> > > Thank you so much for your active participation in the discussion!
> > >
> > > From a theoretical perspective, we've made significant advancements over the works similar to  [[1]]( https://arxiv.org/pdf/2006.08085.pdf ), which exclusively focus on deterministic (and synchronous) algorithms. In  [[1]]( https://arxiv.org/pdf/2006.08085.pdf ), the Laplacian matrix $\Lambda$ is derived from doubly-stochastic gossip matrices, ensuring $\Vert \Lambda\Vert=1$ and guaranteeing that every edge spikes precisely once per communication round (resulting typically in $\chi_2\leq 1$, refer to Prop 3.9 of [[2]]( http://proceedings.mlr.press/v202/nabli23a/nabli23a.pdf ) for an in-depth discussion). Generally, deterministic algorithms establish bounds that depend on the spectral gap $\rho = \Vert \Lambda \Vert \chi_1$, and can be accelerated up to $\sqrt{\rho}$.
> > > Our novel stochastic (and asynchronous) algorithm, $A^2CiD^2$, goes beyond this by providing the potential to achieve a dependency of $\sqrt{\chi_1\chi_2}\leq \sqrt{\rho}$ theoretically (see the star-graph case for which we’re strictly better and refer to Prop 3.9 and Tab. 2 of [[2]]( http://proceedings.mlr.press/v202/nabli23a/nabli23a.pdf )). To our knowledge, we are the pioneers in achieving such communication bounds for an algorithm applicable to Deep Neural Networks.
> > >
> > > From a practical standpoint, synchronous algorithms must *wait for the slowest worker at each step*. Asynchronous algorithms come to the rescue by allowing *each worker to operate at its own pace*. Even though they maintain a similar *total* amount of computations (the training stops when a *total number of grad steps* have been done), slower workers contribute less, while  faster ones contribute more. When compared to the synchronous case, our basic implementation necessitates less communication and less time (table for ImageNet training with 64 workers):
> > >
> > > | method                | Time (min) | # grad (slowest worker) | # grad (fastest worker) |
> > > |-----------------------|------------|-------------------|-------------------|
> > > | AR-SGD (Pytorch DDP)               | 1.7 $10^2$        | 14k            | 14k             |
> > > | AD-PSGD               | 1.5 $10^2$        | 13k             | 14k             |
> > > | AD-PSGD w/ $A^2CiD^2$ | 1.5 $10^2$        | 13k             | 14k            |
> > >
> > >
> > > Consequently, we demonstrate that the **asynchronous scenarios outperform the synchronous one**: $A^2CiD^2$ represents a pioneering, accelerated, and asynchronous algorithm, grounded in both theory and practicality. We firmly believe that this contributes significantly to the advancement of the field.
> > >
> > > [1] *Optimal Complexity in Decentralized Training*, ICML 2021.
> > >
> > > [2] *DADAO: Decoupled Accelerated Decentralized Asynchronous Optimization*, ICML 2023.

---

> > > > ### Comment · Area_Chair_vxtw · 2023-08-16
> > > >
> > > > Thanks for the response. I am still curious the complete comparison of theoretical communication complexity to synchronous algorithms.

---

> > > > > ### Author Response · Authors · 2023-08-16
> > > > >
> > > > > Dear AC,
> > > > >
> > > > > For the sake of simplicity, we will derive the analysis of the communication complexity for the strongly-convex case with full gradient (not SGD), the smooth (and/or stochastic) one being completely analogous.
> > > > >
> > > > > In our paper, we introduce $\Lambda$, the Laplacian matrix weighted with $\lambda_{ij}$, the expected number of times a communication along edge $(i,j) \in \mathcal E$ happens each time unit. Recall that we normalize times so that each node takes one gradient step per time unit in expectation. Thus, *one time unit for us is analogous to one round of computation (one "step") for synchronous methods*. Hence, per time unit, our communication complexity (the expected number of communications) is simply given by $\frac{\text{Tr}(\Lambda)}2$.
> > > > >
> > > > > Synchronous methods such as DeTAG [1], MSDA [2] and OPAPC [3] perform multiple rounds of communications (the Accelerated Gossip procedure in [1, 2, 3]) between rounds of gradient computations by using an inner loop inside their main loop (the one counting "steps"), so that the graph connectivity do not impact the total number of "steps'' necessary to reach $\epsilon$-precision. As **Proposition 3.6** in our paper states $\Vert \bar{x_T} - x^* \Vert^2 = \tilde {\mathcal O} \left( \Vert \bar{x_0} - x^*\Vert^2 e^{- \frac {\mu T}{16 L (1 + \sqrt{\chi_1 \chi_2})}} \right)$, this is analogous to saying $\sqrt{\chi_1[\Lambda] \chi_2[\Lambda]} = \mathcal O (1)$ for our method (i.e., the graph connectivity does not impact the time to converge).
> > > > >
> > > > > Now, let us consider a gossip matrix $W$ as in [1,2,3] (i.e., $W$ is symmetric doubly stochastic) and its Laplacian $\mathcal L =  I_n - W$. Then, using $\Lambda = \sqrt{\chi_1[\mathcal L ] \chi_2[\mathcal L]} \mathcal L$ is sufficient for having $\sqrt{\chi_1[\Lambda] \chi_2[\Lambda]} = \mathcal O (1)$.
> > > > > * **synchronous methods**: between two rounds of computations ("steps"), the number of communication edges used is $\frac{\vert \mathcal E \vert}{\sqrt{1 - \theta}}$ with $\theta = \max \\{ \vert \lambda_2 \vert , \vert \lambda_n \vert \\}$ the eigenvalues of $W$.
> > > > > * **ours**: the number of communication edges used per time unit for our method is $\frac{\text{Tr}(\Lambda)}2 = \frac 1 2 \sqrt{\chi_1[\mathcal L ] \chi_2[\mathcal L]} \text{Tr}( \mathcal L)$.
> > > > >
> > > > > As, in [1,2,3], each communication edge is used at the same rate, we can apply **Lemma 3.3** of [4] stating: $\sqrt{\chi_1[\mathcal L ] \chi_2[\mathcal L]} \text{Tr}( \mathcal L) \leq \sqrt{\Vert \mathcal L \Vert \chi_1[\mathcal L]  (n-1) \vert \mathcal E \vert }$. We have:
> > > > > * $W$ is stochastic: $\Vert \mathcal L \Vert \leq 2$
> > > > > * the graph is connected: $n-1 \leq \vert \mathcal E \vert$
> > > > > * definition of $\chi_1$ and $\theta$: $1 - \theta \leq \frac 1 {\chi_1[\mathcal L]}$
> > > > >
> > > > > Thus, $\sqrt{\chi_1[\mathcal L ] \chi_2[\mathcal L]} \text{Tr}( \mathcal L) \leq \frac { \sqrt{2} \vert \mathcal E \vert}{\sqrt{1 - \theta}}$, which **proves that our communication complexity is better than any accelerated synchronous method**.
> > > > >
> > > > > In some graphs, the difference can be arbitrarily big as shown with the following table:
> > > > >
> > > > > | Method                                  | Comm. complexity per “step”/time unit                                                                                                   | Star Graph | Circle Graph | Complete Graph |
> > > > > |-----------------------------------------|---------------------------------------------------------------------------------------------------------------|------------|--------------|----------------|
> > > > > | Accelerated Synchronous (e.g., [1,2,3]) | $\frac{\vert \mathcal E \vert}{\sqrt{1 - \theta}}$                                                            | $n^{3/2}$  | $n^2$        | $n^2$          |
> > > > > | Ours                                    | $ \sqrt{\chi_1\chi_2} \text{Tr}( \mathcal L)$ | $n$        | $n^2$        | $n$            |
> > > > >
> > > > >
> > > > > We will add those elements to our paper, and we really hope that we answered in a satisfactory manner to your concerns about the optimality of stochastic (asynchronous) algorithms. Note that the fact that stochastic optimization algorithms are faster than their deterministic counterparts is something well known in centralized settings (e.g., [5]).
> > > > >
> > > > > [1] *Optimal Complexity in Decentralized Training*, ICML 2021.
> > > > >
> > > > > [2] *Optimal Algorithms for Smooth and Strongly Convex Distributed Optimization in Networks*, ICML 2017
> > > > >
> > > > > [3] *Optimal and Practical Algorithms for Smooth and Strongly Convex Decentralized Optimization*, NeurIPS 2020.
> > > > >
> > > > > [4] *DADAO: Decoupled Accelerated Decentralized Asynchronous Optimization*, ICML 2023.
> > > > >
> > > > > [5] *Tight Complexity Bounds for Optimizing Composite Objectives*, NIPS 2016.

---

> > ### Comment · Reviewer_o6Xd · 2023-08-20
> > **Thanks for the response**
> >
> > Thank you for the response.
> >
> > The authors have addressed most of my concerns. Hence, I raise my score by one level.
> > The issue about experiment has not been addressed. AD-PSGD is relatively outdated. More advanced baselines should be adopted for comparison, such as DADAO (ICML 2023) and other methods cited in the reference list.
> >
> > DADAO: Decoupled Accelerated Decentralized Asynchronous Optimization, ICML 2023.

---

> > > ### Author Response · Authors · 2023-08-20
> > > **AD-PSGD is the state-of-the-art baseline in asynchronous decentralized DNN training**
> > >
> > > Unfortunately, DADAO is designed and evaluated for the convex case and isn't suitable for training DNNs as it relies on the saddle points of a Lagrangian, which has no equivalent in Deep Learning. Indeed our preliminary experiments for this paper found DADAO obtained poor performance for DNNs in practice and it was excluded for further consideration. To the best of our knowledge AD-PSGD stands as the state-of-the-art baseline for asynchronous decentralized DNN training algorithms (see also reply to reviewer fkqi), thus we ask the reviewer to reconsider their assessment regarding the strengths of the baselines.

---

### Official Review · Reviewer_fkqi · 2023-07-28

**Soundness:** 2 fair
**Presentation:** 3 good
**Contribution:** 2 fair
**Rating:** 5
**Confidence:** 3

**Summary:**

This work introduces a new method for decentralized optimization that leverages the notion of continuous momentum to speed up its convergence. The method is justified with theoretical analysis and large-scale experiments on the ImageNet dataset.

**Strengths:**

* The work studies an important problem of distributed training over communication-constrained networks
* The results of authors have both theoretical justification and empirical validation in a large-scale setting
* Overall, the approach is clearly explained and the contributions are easy to understand

**Weaknesses:**

* The primary disadvantage of the proposed method is its memory overhead. Having an additional copy of model parameters on each worker is quite expensive for models where the memory footprint of the parameters dominates that of the activations (in particular, for Transformer models). As a result, it might be quite difficult to apply A$^2$CiD$^2$ to models beside convolutional networks (for example, to train modern language models), which are a highly popular area of application for distributed training
* I feel that the statement in L241 needs to be clarified. All-Reduce methods indeed require more *connections* with the growth in the number of workers, but the total amount of bandwidth is asymptotically independent of the network size for methods like Ring All-Reduce
* From my understanding, momentum acceleration for decentralized DL has already been studied in the past (e.g., [1], which has been cited in the submission), although the underlying frameworks are clearly different. I think this work needs to more explicitly distinguish their approach from the QG-DSGDm and other results (e.g., [2]), and ideally include those methods as an additional baseline on top of AD-PSGD

[1] Quasi-Global Momentum: Accelerating Decentralized Deep Learning on Heterogeneous Data. Tao Lin, Sai Praneeth Karimireddy, Sebastian U. Stich, Martin Jaggi. ICML 2021

[2] SlowMo: Improving Communication-Efficient Distributed SGD with Slow Momentum. Jianyu Wang, Vinayak Tantia, Nicolas Ballas, Michael Rabbat. ICLR 2020

**Questions:**

* In L197, what is the connection speed between the nodes?
* What was your motivation for choosing the ring graph topology in the experiments? Is this topology frequently used in practice for peer-to-peer networks?
* Is it correct that you consider standard All-Reduce to be a centralized method? (e.g., L80) From my understanding, there is no central worker involved when aggregating updates with this family of algorithms

**Limitations:**

I did not see any explicit discussion of limitations of the work: I would be happy if they mentioned the applicability of their method to models with a larger parameter count (for example, Transformers with hundreds of millions or billions of parameters).

---

> ### Author Rebuttal · Authors · 2023-08-10
>
> We thank reviewer fkqi for acknowledging the importance of the problem studied and remarking that our theoretical and practical contributions are clear. Among those, we would like to emphasize that obtaining an accelerated rate of communication in the **asynchronous** setting is non-trivial, and mostly ignored in the literature. We thus consider our work to be an important contribution to the field.
>
> **Weaknesses:**
>
> *1) Memory overhead of an additional copy of the model dominates the activations memory*
>
>
> We respectfully disagree with this statement. As their activations memory usually scales with the *squared of the sequence length*, the activations memory requirements of Transformers dominates the memory footprint, which led to many work to consider new methods to reduce this activation cost (see e.g,  [ [ 1, ]( https://arxiv.org/pdf/2111.11124.pdf ) [ 2 ]( https://arxiv.org/pdf/2205.05198.pdf ) ] ). We emphasize that our method only uses the second set of parameters as a *"memory bank"*: our second "model" does not perform any computation (no forward nor backward), and only one optimizer is needed (see eq. 4 and l.9 of Algorithm 1). Therefore, our method needs the memory requirement for only one set of activations and gradients. Thus, it is virtually at no cost in terms of speed (no additional heavy compute), and no cost in terms of memory (the memory for activations dominating the storing of parameters).
> Finally, remark that *any* algorithm that uses SGD with momentum is also doubling the number of parameters in memory: that is the cost of storing the momentum variable in the optimizer. Thus, our method is completely analogous to a standard momentum in that sense.
>
> *2) Bandwidth considerations of All-reduce and Ring reduce*
>
> Thank you for pointing that out, we will clarify our statement. We are indeed interested in reducing the latency induced by each "communication act" between workers, and as stated in the Ring All-Reduce paper [ [3] ](https://www.cs.fsu.edu/~xyuan/paper/09jpdc.pdf ): *"One limitation of the proposed algorithm is that it is only optimal in the bandwidth term, but not the latency term: the number of communication rounds is proportional to the number of processes."* An orthogonal (but crucial) line of work [ [8](https://arxiv.org/pdf/1610.02132.pdf ), [9](https://arxiv.org/pdf/1802.04434.pdf) ] is indeed to consider lowering the bandwidth in addition to the total number of communications, using compression schemes for example, to which our method can be independently combined with. Moreover, we stress that Ring All Reduce is **synchronous**, adding undesirable barriers that we remove by focusing on asynchronous methods.
>
> *3) Distinction from QG-SGD, SlowMo*
>
> * QG-DSGDm [ [4] ](https://arxiv.org/pdf/2102.04761.pdf ) introduces a momentum to lower the complexity of **synchronous rounds of computations-communications** in the **heterogeneous** setting. We are rather interested in explicitly lowering the **communication complexity** in the **asynchronous** and **homogeneous** setting. Moreover, in addition to being synchronous, their method requires gradients to be computed **before** communicating, meaning the latencies for computations and communications are **added**, whereas, being decoupled, our method allows running both processes **in parallel.**
>
> * SlowMo [ [5] ]( https://arxiv.org/pdf/1910.00643.pdf ) is also a **synchronous** algorithm.
>
> We must emphasize that we study an **asynchronous** algorithm. This brings many practical advantages in the large-scale distributed setting, but also leads to specific technical challenges (for example, the Chebyshev scheme widely used to accelerate communications in decentralized methods is, by nature, synchronous and sequential). As such, the closest method to which we can compare is AD-PSGD [ [6] ]( http://proceedings.mlr.press/v80/lian18a/lian18a.pdf ), which we have done. *We stress that no other existing decentralized asynchronous method displays accelerated rates of communications in the training of deep neural networks*.
>
> **Questions:**
>
> * The cluster we used has an *Omni-PAth interconnection network 100 Gb/s*. We will add this point in the main text, thanks.
> * Our work aims at studying the impact of the network’s connectivity  (as measured by $\chi_1, \chi_2$) on the communication complexity in the decentralized training setup, and showing that $A^2CiD^2$ allows to reduce it. As the ring graph corresponds to one of the worst case settings, it is standard to use it in the decentralized literature, see e.g  [ [4, ](https://arxiv.org/pdf/2102.04761.pdf ) [ 7]( https://arxiv.org/pdf/1705.09056.pdf) ].
> * While it is technically true that there is no central worker performing the averaging in the *implementation* of modern All-Reduce methods, we still denote them as centralized as they require the computation of a *global* variable using information from *all* the workers. Put simply, although there are multiple ways to implement it, the requirement for waiting barriers and synchronized communication rounds corresponds exactly to a centralized framework.
>
> **Limitations:**
>
> We emphasize that our evaluation is standard in the literature (see [ [10](https://arxiv.org/pdf/1808.07217.pdf), [11](https://arxiv.org/pdf/1811.10792.pdf) , [6]( http://proceedings.mlr.press/v80/lian18a/lian18a.pdf ) ]). While we agree that verifying experimentally that our method scales to training models with billions of parameters would be of interest, we are unfortunately not in the material capacity to do so, lacking the right computing infrastructures: we must stress that we work in **academia** with a **publicly funded cluster**. The open-source release of our code is planned to allow such experiments for other actors with more compute resources. However, we will aso report additional experiments with Vision Transformers shortly (experiments still ongoing at the time of writing), but we expect a similar behavior for these models.

---

### Author Rebuttal · Authors · 2023-08-10

We thank reviewers for recognizing that our method accelerates previous state of the art **asynchronous decentralized** methods both theoretically and empirically (reviewers fkqi, o6Xd, fQ1m) which is a good step towards addressing common challenges of large scale training of deep neural networks (reviewer LKYk). We would now like to address frequently raised comments, which we believe will clarify any previous concerns:

* **The memory footprint of our method is not high.**
Our method is completely analogous to a standard momentum: **any** algorithm that uses SGD with momentum is also doubling the number of parameters in memory: that is the cost of storing the momentum variable in the optimizer. As for any momentum term, no additional heavy computation is performed with these parameters (no forward nor backward), making the cost in time and memory negligible for large models. For instance, *our method (SGD+momentum+$A^2CiD^2$) has the same memory footprint as Adam* (as it has 2 momentum variables).

* **Our academic cluster is not able to train models with billions of parameters, and our experiments are standard.**
We are *academic* researchers using a *publicly funded* cluster: we were surprised that reviewers requested us to train GPT-like models.
Moreover, training ResNets on CIFAR10 and ImageNet is standard in the decentralized training literature (see [ [1](https://arxiv.org/pdf/1808.07217.pdf), [2](https://arxiv.org/pdf/1811.10792.pdf) , [3]( http://proceedings.mlr.press/v80/lian18a/lian18a.pdf ) ]).

* **Cluster details.**
All our experiments are done on a cluster with 8 A100 per node using an Omni-PAth interconnection network 100 Gb/s. In all our experiments, one worker amounts to one GPU (and not one node). We will add these details in our paper.

* **A general remark on Communication Acceleration.**
Before this work, communication acceleration - enhancing the communication rate from $\gamma$ (the spectral gap of the matrix) to $\sqrt{\gamma}$ - was (almost) exclusively achieved via Chebychev acceleration, in the context of training DNNs. Thus, most previous studies don't dwell on this, as it simply corresponds to replacing a gossip matrix $W$ with $P(W)$ where $P$ is the corresponding Chebychev polynomial. However, Chebychev acceleration is inherently synchronous and sequential. Thus, such a process is absolutely undesirable in an asynchronous environment, underscoring the significance of our contribution. *It's essential for reviewers to grasp this distinction and recognize that, prior to our work, standard gossip algorithms that simply perform an averaging between node parameter values were the norm.* Thus, we believe our work to be an important contribution to the litterature of asynchronous training procedures for DNNs.

---

### Decision · Program_Chairs · 2023-09-21

**Decision:**

Accept (poster)

**Comment:**

This paper studies the asynchronous parallel algorithm in the decentralized setting and proposes a random communication based approach. Both theoretical analysis and empirical evaluations are provided. While the theoretical results look reasonable, the empirical study is not sufficient to validate the proposed algorithm, which is an obvious weakness of this paper as pointed by multiple reviewers. We finally decide to accept this borderline paper given its contribution in theory, but strongly suggest this paper to improve the experiment section.

Specific suggestions include:

- experiment: study the practical performance for the proposed algorithm under different scenarios to validate the theory. In particular, we want to know when it outperforms the synchronous algorithms and AD-PSGD. More solid comparison to the synchronous approaches need to be included to justify the advantage of synchronization. The comparison to All-reduce (the most classic synchronous algorithm) need to be solid and fair. Authors may consider using existing distributed learning framework to implement the designed designed algorithm, avoid complicated and nontrivial system optimization, e.g., [1]

- theory: provide more comprehensive comparison to the synchronous algorithms for validating the tightness of the proposed analysis. The comparison to the asynchronous algorithm in the centralized setting needs to be compared, e.g., [2]. We believe this can be achievable in a relatively short term based on the author's response.

- algorithm: some other related decentralized / asynchronous algorithm comparison needs to be included, e.g. [3-12]

References:

[1] BAGUA: Scaling up Distributed Learning with System Relaxations  https://github.com/BaguaSys/bagua

[2] Distributed Learning Systems with First-Order Methods: An Introduction

[3] Asynchronous Gradient-Push https://arxiv.org/pdf/1803.08950.pdf

[4] Optimal Complexity in Decentralized Training, ICML 2021

[5] Optimal Algorithms for Smooth and Strongly Convex Distributed Optimization in Networks, ICML 2017

[6] Optimal and Practical Algorithms for Smooth and Strongly Convex Decentralized Optimization, NeurIPS 2020.

[7] DADAO: Decoupled Accelerated Decentralized Asynchronous Optimization, ICML 2023.

[8] A Unified Theory of Decentralized SGD with Changing Topology and Local Updates, ICML 2020

[9] Stochastic Gradient Push for Distributed Deep Learning, ICML 2019

[10] Don’t use large mini-batches, use local SGD, ICLR 2020

[11] Asynchronous Decentralized Parallel Stochastic Gradient Descent, ICML 2018

[12] Consensus control for decentralized deep learning, ICML 2021